# ACCELERATING AUTO-REGRESSIVE TEXT-TO-IMAGE GENERATION WITH TRAINING-FREE SPECULATIVE JACOBI DECODING

**Yao Teng[1]   Han Shi[2]   Xian Liu[3]   Xuefei Ning[4]**
**Guohao Dai[5,6]   Yu Wang[4]   Zhenguo Li[2]   Xihui Liu[1]***
[1]The University of Hong Kong   [2]Huawei Noah's Ark Lab   [3]CUHK
[4]Tsinghua University   [5]Shanghai Jiao Tong University   [6]Infinigence AI

## ABSTRACT

The current large auto-regressive models can generate high-quality, high-resolution images, but these models require hundreds or even thousands of steps of next-token prediction during inference, resulting in substantial time consumption. In existing studies, Jacobi decoding, an iterative parallel decoding algorithm, has been used to accelerate the auto-regressive generation and can be executed without training. However, the Jacobi decoding relies on a deterministic criterion to determine the convergence of iterations. Thus, it works for greedy decoding but is incompatible with sampling-based decoding which is crucial for visual quality and diversity in the current auto-regressive text-to-image generation. In this paper, we propose a training-free probabilistic parallel decoding algorithm, **Speculative Jacobi Decoding (SJD)**, to accelerate auto-regressive text-to-image generation. By introducing a probabilistic convergence criterion, our SJD accelerates the inference of auto-regressive text-to-image generation while maintaining the randomness in sampling-based token decoding and allowing the model to generate diverse images. Specifically, SJD facilitates the model to predict multiple tokens at each step and accepts tokens based on the probabilistic criterion, enabling the model to generate images with fewer steps than the conventional next-token-prediction paradigm. We also investigate the token initialization strategies that leverage the spatial locality of visual data to further improve the acceleration ratio under specific scenarios. We conduct experiments for our proposed SJD on multiple auto-regressive text-to-image generation models, showing the effectiveness of model acceleration without sacrificing the visual quality. The code of our work is available here: `https://github.com/tyshiwo1/Accelerating-T2I-AR-with-SJD/`.

## 1  INTRODUCTION

Auto-regressive models enable generative tasks by performing next-token prediction, which is widely used in multiple domains such as the language (Bubeck et al., 2023), image (Yu et al., 2022), and video (Kondratyuk et al., 2023; Wang et al., 2024b) generation. Notably, auto-regressive text-to-image generation models (Ding et al., 2021; Ramesh et al., 2021; Yu et al., 2022) have shown promising results in generating high-quality images. Auto-regressive text-to-image generation models have better potential in scalability and pave the way for native multi-modal models (Team, 2024). However, the auto-regressive paradigm creates high latency during inference because it necessitates the decoding of tokens in a sequential, token-by-token manner. Therefore, the models have to sequentially go through hundreds or even thousands of forward passes to generate a single image. Unlike diffusion models of which the inference acceleration methods have been extensively investigated (Song et al., 2023; Luo et al., 2023; Yin et al., 2024b), there has been limited previous work exploring the acceleration of auto-regressive text-to-image generation models. Moreover, those auto-regressive models that are capable of text-to-image generation typically have several billions of parameters, making the common training-based generative model acceleration techniques such as self-consistency

---

*Corresponding Author

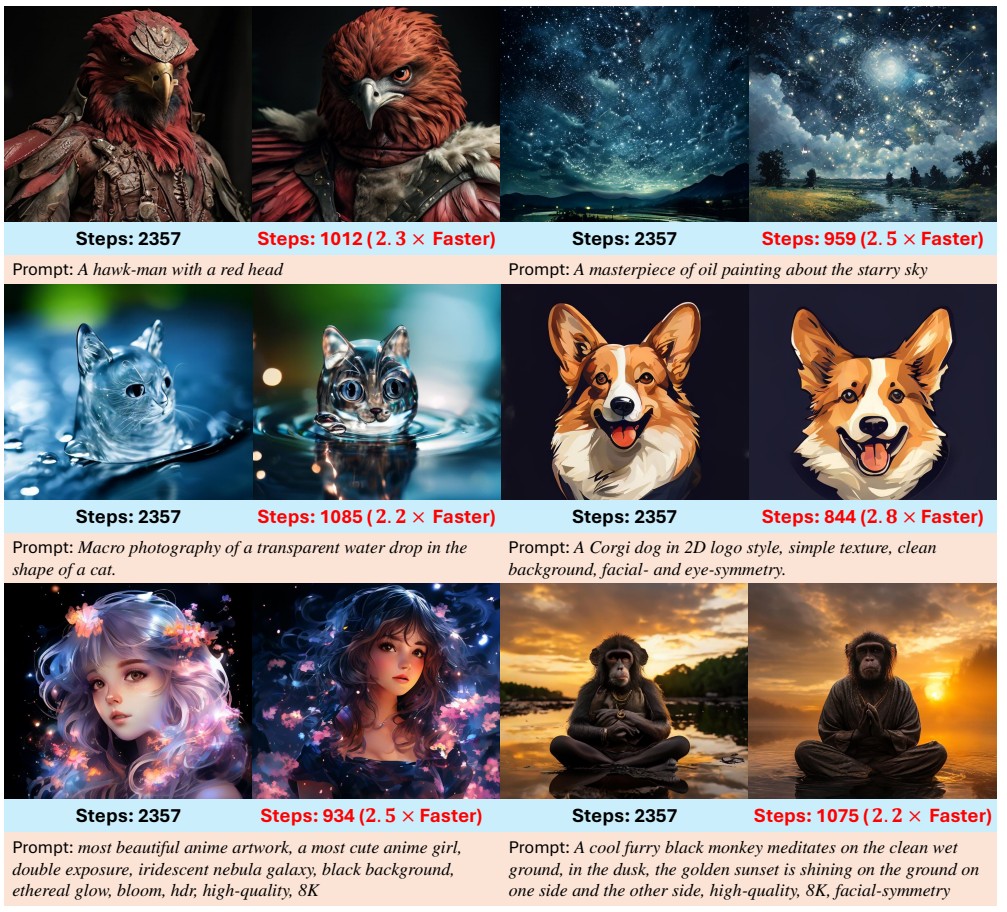

Figure 1: We propose Speculative Jacobi Decoding, a training-free multi-token prediction algorithm, to accelerate auto-regressive text-to-image generation by reducing the number of model forward passes (denoted as `steps`) during inference. We perform our algorithm on Lumina-mGPT, and the reduced steps are marked in red. The original steps are marked in black.

distillation computationally expensive (Kou et al., 2024). Therefore, our work aims to accelerate the auto-regressive text-to-image generation models in a training-free manner.

An intuitive approach is to enable the auto-regressive models to decode multiple tokens in parallel within a forward pass. In the early research on auto-regressive image generation, Jacobi decoding (Ortega & Rheinboldt, 2000) has been employed to achieve this objective (Song et al., 2021). Jacobi decoding is an iterative algorithm starting from a sequence of randomly initialized tokens, and this algorithm can be executed directly on pre-trained auto-regressive models in a training-free way. In each Jacobi iteration, the model performs a single forward pass on the input sequence with a causal mask, thus decoding tokens in parallel. The decoded tokens would converge after multiple iterations of parallel decoding. The criterion for this convergence is defined as follows: the difference between the values of decoded tokens remains within a sufficiently small threshold over two consecutive iterations. Since the number of iterations required for convergence is typically smaller than the sequence length and the parallel forward pass runs fast in GPUs, the generation can be accelerated with Jacobi Decoding.

However, Jacobi decoding faces significant challenges when applied to recent auto-regressive text-to-image generation models. We observe that the recent auto-regressive text-to-image generation models (Liu et al., 2024b; Chern et al., 2024; Sun et al., 2024a) greatly rely on sampling-based decoding with high randomness to generate diverse images. We present the generated images using top-$K$ sampling with various $K$ values, where a larger $K$ indicates higher randomness. As shown in Fig. 2, the model with high randomness in sampling generates images with diverse and high-fidelity details and structures, whereas it outputs monotonous or even incomprehensible images with greedy decoding. Unfortunately, Jacobi decoding with the deterministic criterion of convergence is

incompatible with such highly random sampling (analyzed in Sec. 5.4), *i.e.*, and it cannot accelerate the inference given such sampling decoding.

In this work, we propose to use a *probabilistic Jacobi decoding* algorithm to accelerate the inference of auto-regressive text-to-image generation models and to support the sampling decoding methods for those models. We observe that the acceleration of Jacobi decoding relies on the assumption that multiple consecutive tokens can be correctly decoded in each Jacobi iteration (shown by **green stepped area** in Fig. 3). Similar ideas have been applied in other probabilistic algorithms for accelerating the decoding of large language models. For example, in speculative sampling (Leviathan et al., 2023; Chen et al., 2023), an additional small model is trained for rapidly generating draft sequences, and then the large language model probabilistically accepts a subset of draft tokens from left to right. Drawing from the above analysis, in this paper, we directly advance the deterministic Jacobi decoding into a probabilistic algorithm, coined as Speculative Jacobi Decoding (SJD). Our method allows the auto-regressive text-to-image generation models to decode *multiple tokens* within *one* forward pass in a *training-free* manner. In SJD, the model computes the conditional probability for a sequence of *draft tokens* with a single forward pass. Then, we define a probabilistic criterion to determine which draft tokens to accept, from left to right. The accepted tokens are appended to the fixed pre-filling sequence. The remaining tokens are concatenated with a set of newly initialized tokens, serving as the *draft tokens* for the next decoding iteration. Our SJD accelerates the inference of auto-regressive text-to-image generation models without requiring additional training or tuning of separate modules. Moreover, we propose the spatial locality-aware token initialization strategy to accelerate the generation process further.

We perform quantitative and qualitative experiments to demonstrate the effectiveness of our method. Results show that our method can accelerate several auto-regressive text-to-image generation models without sacrificing the quality of generated images. For example, it can accelerate Anole (Chern et al., 2024) and Lumina-mGPT (Liu et al., 2024b) by about $2\times$ with almost no loss in visual quality. Moreover, the acceleration ratio can be beyond $3\times$ in certain scenarios containing simple patterns.

To the best of our knowledge, SJD is the first method for accelerating the inference of auto-regressive text-to-image models that rely on sampling decoding. We summarize our contributions as follows:

- We propose a new probabilistic multi-token decoding algorithm, coined as **Speculative Jacobi Decoding (SJD)**. By improving the previous Jacobi decoding with a probabilistic criterion for token acceptance, we can accelerate the recent auto-regressive text-to-image generation models that rely heavily on random token samplers.

- Compared with previous Speculative Decoding to accelerate language models, our approach is training-free and does not require training an extra model to predict draft tokens.

- Experiments demonstrate that our method can accelerate auto-regressive text-to-image generation by around $2\times$ with almost no sacrifice in visual quality.

## 2 RELATED WORK

**Auto-regressive image generation.** Auto-regressive image generation models have two features: *next-token-prediction* and *discrete image tokenization*. Early works including PixelCNNs (Van den Oord et al., 2016; Salimans et al., 2017) and PixelSNAIL (Chen et al., 2018) use the auto-regressive strategy to model the image generation with the convolutional neural networks on the discretized pixels. These works generate pixels in the raster-scan ordering or the zigzag ordering. DALL-E (Ramesh et al., 2021) and CogView (Ding et al., 2021) pave the way for the pipeline of the auto-regressive image generation: A discrete autoencoder compresses RGB images into image tokens and a large auto-regressive model makes predictions based on these image tokens. Parti (Yu et al., 2022) uses a transformer encoder (Vaswani et al., 2017) to provide the textual features for the auto-regressive model to perform the next image token prediction, thereby achieving text-to-image generation. LlamaGen (Sun et al., 2024a) acts as a class-to-image auto-regressive baseline on ImageNet dataset (Deng et al., 2009). MARS (He et al., 2024) performs multi-modal generation with a mixture of auto-regressive models, where its image model is initialized with the pre-trained large language model and is fine-tuned to perform image generation. Chameleon (Team, 2024) aims to unify all multi-modal tasks with discrete tokens and perform the next token prediction on these tokens with a large auto-regressive model. Lumina-mGPT (Liu et al., 2024b) and Anole (Chern et al.,

2024) fine-tune Chameleon for better text-to-image generation. In this paper, we conduct experiments mainly on Lumina-mGPT and Anole to verify the effectiveness of our method.

**Acceleration of image generation models.** The iterative image generation requires acceleration. For instance, the diffusion model, originally trained on a denoising trajectory with one thousand steps, has been accelerated to perform inference using just dozens or even a few steps. Given that the diffusion model has emerged as a leading approach in text-to-image generation (OpenAI, 2023; Rombach et al., 2022; Esser et al., 2024), most acceleration methods in image generation are built upon it. Many acceleration methods focus on shortening the denoising trajectory by distillation technique (Salimans & Ho, 2022; Song et al., 2023; Wang et al., 2024a; Kim et al., 2023; Xu et al., 2024; Yin et al., 2024b;a) while some other studies focus on reducing the computational complexity (Yuan et al., 2024; Zhao et al., 2024; Ma et al., 2024). In contrast to the diffusion model, acceleration methods for auto-regressive image generation have not been extensively explored, primarily due to the absence of powerful base models. Jacobi decoding is applied to PixelCNNs for inference acceleration in the early research (Song et al., 2021), yet it lacks a careful design for the random token sampling, significantly impacting its acceleration on current auto-regressive models. In this paper, we enhance Jacobi decoding to be compatible with this random sampling. Also, the inference process of each iteration in our method is similar to that of non-auto-regressive models (Chang et al., 2022; Tian et al., 2024; Li et al., 2024b). Nevertheless, unlike these models, our approach only modifies the inference schedule of pre-trained auto-regressive models instead of training a separate non-auto-regressive model, thereby preserving the performance and scalability of the auto-regressive models.

**Acceleration of language models.** Different from image generation, the auto-regressive paradigm prevails in language processing. A lot of works (Zhou et al., 2024; Devoto et al., 2024; Liu et al., 2024a;c; Yang et al., 2024a; DeepSeek-AI, 2024; Zhang et al., 2024; Fu et al., 2024a; Li et al., 2024a) focus on compressing the models by weight pruning, activation sparsification, quantization, factorization, but the paradigm of token-by-token prediction remains unchanged. There are also works fine-tuning the auto-regressive models to predict multiple tokens in parallel with several decoding heads (Gloeckle et al., 2024). However, these works require more memory to load these additional heads in GPUs. The speculative sampling (Leviathan et al., 2023; Chen et al., 2023; Li et al., 2024c; Sun et al., 2024b) uses a small language model to assist the large language model in sequence generation. This model is trained on the same domain as the large model and is small enough for faster generation. It first generates a sequence with its own inference paradigm. Then, the large model verifies and samples only one prefix of this sequence to serve as part of the final output by executing a single forward pass. The verification phase is well-designed to guarantee that each sampled token theoretically satisfies the conditional probability parameterized by the large model. Jacobi decoding (Song et al., 2021; Santilli et al., 2023) allows the model to iteratively decode multiple tokens in fewer steps than the token counts with the deterministic greedy sampling but without auxiliary modules. CLLM (Kou et al., 2024) fine-tunes the collected Jacobi trajectories into large language models for acceleration. Lookahead Decoding (Fu et al., 2024b) adapts training-free Jacobi decoding in large language models by using a pool of $n$-grams obtained via Jacobi iterations with greedy sampling. In this work, we directly adapt the probabilistic verification of speculative sampling into Jacobi decoding to advance it into a probabilistic algorithm without any additional auxiliary designs and training.

## 3 PRELIMINARIES

### 3.1 AUTO-REGRESSIVE TEXT-TO-IMAGE GENERATION

The auto-regressive text-to-image generation models are composed of three components: a discrete image tokenizer that encodes images into discrete tokens, an auto-regressive transformer-based generator that generates discrete image tokens with next-token-prediction conditioned on the text prompts, and an image decoder that decodes the predicted image tokens to the images in pixel space. The most time-consuming component for auto-regressive text-to-image generation is the auto-regressive transformer, and our work aims at accelerating the inference of the auto-regressive transformer to predict discrete image tokens based on text prompts.

During each inference step of the auto-regressive transformer, the model predicts the probability distribution of the next token over the entire vocabulary of the tokenizer (implemented through a softmax classifier) and then samples from this distribution to generate the token. Specifically, given a

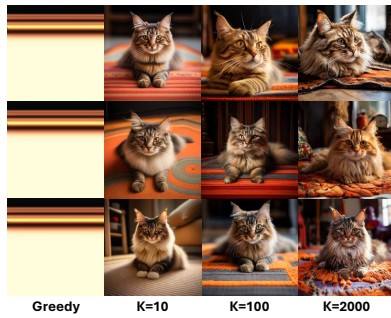

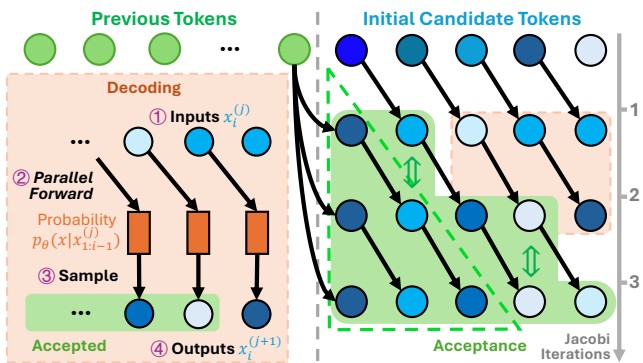

Figure 2: The results of the greedy decoding (no randomness), top-10, top-100, and top-2000 sampling (high randomness) of Lumina-mGPT (Liu et al., 2024b). Each row presents the images generated with the same *random seeds*.

Figure 3: The pipeline of the vanilla Jacobi decoding on an auto-regressive model. The prediction with sampling is performed in parallel at each Jacobi *iteration*. We use different shades of blue to indicate the differences between the tokens that have not been accepted.

sequence of pre-filled or already decoded tokens $(x_1, x_2, \cdots, x_i)$, the auto-regressive model predicts a categorical distribution $p_\theta(x|\boldsymbol{x}_{1:i})$, where we denote the input token sequence $(x_1, x_2, \cdots, x_i)$ as $\boldsymbol{x}_{1:i}$ for simplicity, $\theta$ denotes the auto-regressive model parameters, and $x$ is the random variable representing the next token (category). Then, a token is sampled according to $p_\theta(x|\boldsymbol{x}_{1:i})$, treated as $x_{i+1}$, and is subsequently appended to $(x_1, x_2, \cdots, x_i)$ for the next decoding step. In text-to-image auto-regressive generation, the above process starts with a sequence of text tokens and a special token to represent the beginning of image token prediction. To facilitate the generation of diverse images, top-$K$ sampling is commonly employed as the token sampling strategy for text-to-image generation.

## 3.2 JACOBI DECODING

Jacobi decoding deems the auto-regressive inference as a process of solving the fixed point of a nonlinear equation in a triangular system (Song et al., 2021). This decoding algorithm iteratively performs multi-token decoding and can be executed without fine-tuning or auxiliary modules. We show the specific process of decoding one sequence of tokens in Fig. 3. First, given the previously pre-filled or decoded tokens, we randomly initialize a sequence of candidate tokens. Then, *in each iteration*, we execute one forward pass of the auto-regressive model for all the candidate tokens with a causal mask. The predicted probabilities then generate the tokens via *greedy sampling*, and these sampled tokens are taken as the inputs of the next iteration. This process can be formulated as: $x_i^{(j+1)} = \arg\max_x p_\theta(x|\boldsymbol{x}_{1:i-1}^{(j)})$, where $i$ denotes the token index and $j$ denotes the iteration index. The Jacobi decoding process continues iterating until the convergence is reached, as determined by a deterministic criterion where these tokens remain unchanged between consecutive iterations.

**Discussion.** The acceleration of Jacobi decoding derives from an assumption that multiple tokens can be correctly decoded within one forward pass in Jacobi iteration. Fig. 3 illustrates this scenario, where the accepted tokens (**green stepped area**) extend beyond the dashed **green triangle outline**. Specifically, the model accepts *two consecutive tokens* after the *first* Jacobi iteration. Thus, it can generate at least four tokens through three forward passes. In the worst case, only three tokens can be generated through three forward passes (Song et al., 2021). Note that the number of forward passes in the worst case of Jacobi decoding is equal to that in the original auto-regressive case.

## 4 SPECULATIVE JACOBI DECODING

**Analysis.** The vanilla Jacobi decoding incorporates a deterministic criterion for determining the convergence, which works well with greedy sampling (no randomness) in language models (Fu et al., 2024b; Kou et al., 2024). In contrast, in auto-regressive text-to-image generation, randomness plays a crucial role in the sampling-based decoding process, *i.e.*, higher randomness corresponds to highly diverse details and structures in the generated images. As shown in Fig. 2, we use the text prompt "a cat on a mat" to generate images with different sampling strategies including greedy decoding and

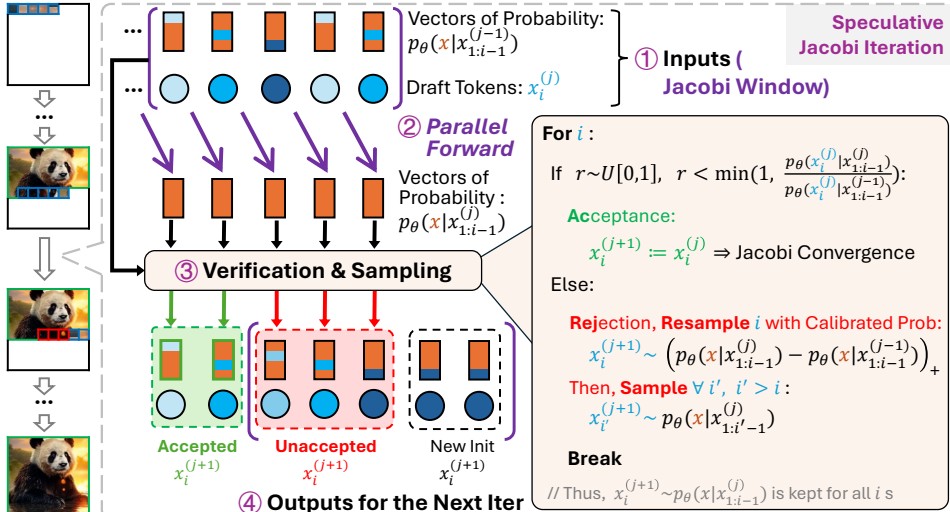

Figure 4: **Overview of one iteration of our speculative Jacobi decoding (SJD).** First, a sequence of draft tokens and the corresponding probabilities are taken as the inputs. Second, we perform a forward pass with the auto-regressive model on the draft tokens, obtaining the probabilities of these tokens. Third, we perform the verification according to these two types of probabilities, accepting a subset of tokens and (re-)sampling the remaining tokens. Last, the accepted tokens are appended to the pre-filling tokens and fixed, while the resampled tokens, along with newly initialized tokens, will serve as the draft tokens for the next iteration.

top-$K$ decoding with different values of $K$. We observe that the generated images would contain more details and diverse structures as the $K$ increases. The greedy decoding ($K = 1$ with no equal probabilities) leads to suboptimal performance with low quality and no diversity in generated images. Therefore, random sampling-based decoding is important to image generation, but the original Jacobi decoding is incompatible with such randomness in sampling.

To address the aforementioned issue, we advance the deterministic Jacobi iteration into a new training-free probabilistic parallel decoding algorithm, inspired by speculative sampling (Leviathan et al., 2023). Specifically, in each iteration, we decode multiple tokens in parallel and utilize a probabilistic criterion to accept multiple decoded tokens from the outputs of the previous iteration. Moreover, to reduce the number of iterations required for inference, we propose a new image token initialization strategy incorporating spatial priors.

## 4.1 SPECULATIVE JACOBI ITERATION

After pre-filling the tokens of text prompts, we perform speculative Jacobi decoding for the image tokens. Acknowledging the computational expense of decoding all image tokens simultaneously, we decode multiple tokens per iteration in a sliding-window manner, termed the Jacobi window. Our method starts with a sequence of initialized candidate tokens, the length of which corresponds to the window size. During each decoding iteration, we predict the token probabilities for the draft token sequence within the current window. Then, a subset of these tokens is accepted based on a probabilistic criterion, and these accepted tokens are added to the fixed pre-filling tokens for the next iteration. The remaining unaccepted tokens are resampled for the next iteration. In the next iteration, the Jacobi windows slides to include the unaccepted tokens from the previous iteration plus some newly initialized tokens to maintain the Jacobi window size during decoding. The process of our iteration is illustrated in Fig. 4. Assuming that we have pre-filled or accepted $n$ tokens and the Jacobi window size is $W$, we would decode the next $W$ tokens. The iteration can be described as follows:

**Step 1**: At the $j$-th iteration, we have the input tokens that are either predicted (but not accepted) in the previous $(j-1)$-th iteration or newly initialized (Sec. 4.2). These tokens serve as the *draft tokens* in this iteration, denoted as $(x_n^{(j)}, x_{n+1}^{(j)}, \cdots, x_{n+W-1}^{(j)})$. We denote the probability corresponding to the draft token $x_i^{(j)}$ as $p_\theta(x|\boldsymbol{x}_{1:i-1}^{(j-1)})$. This probability is *set to be conditioned on the input tokens of the previous iteration* (**Step 3** ensures this setting).

**Step 2**: We execute *a single forward pass* of the auto-regressive model to obtain the conditional probability for the draft tokens in parallel. The probability for $x_i^{(j)}$ is denoted as $p_\theta(x|\boldsymbol{x}_{1:i-1}^{(j)})$.

**Step 3**: We conduct the speculative verification between the conditional probability given the draft tokens from the previous iteration, $p_\theta(x|\boldsymbol{x}_{1:i-1}^{(j-1)})$, and the conditional probability from the current iteration, $p_\theta(x|\boldsymbol{x}_{1:i-1}^{(j)})$. In this verification process, we scan the draft sequence from left to right and determine the acceptance of each token based on a probabilistic threshold. We measure the ratio of the probability conditioned on the draft tokens in the current iteration to that conditioned on the tokens from the previous iteration. Intuitively, this ratio measures how well the token is decoded from the previous iteration (the draft token) and whether further decoding is necessary. The acceptance criterion of a token $x_i^{(j)}$ can be formulated as follows:

$$x_i^{(j+1)} \leftarrow x_i^{(j)} \text{ if } r \sim \mathcal{U}[0,1], \ r < \min\left(1, \frac{p_\theta(x_i^{(j)}|\boldsymbol{x}_{1:i-1}^{(j)})}{p_\theta(x_i^{(j)}|\boldsymbol{x}_{1:i-1}^{(j-1)})}\right), \tag{1}$$

where $r$ is a random variable and $\mathcal{U}[0,1]$ represents a uniform distribution between $0$ and $1$. If a token meets the above criterion, it is accepted and appended to the pre-filling token sequence for the next iteration, *i.e.*, $x_i^{(j')} = x_i^{(j)} \ \forall j' > j$. After accepting one token, we continue this scan until a token is rejected. If the token $x_i^{(j)}$ is rejected, *i.e.*, the inequality in Equ. (1) is not true, we resample a new token by a calibrated distribution:

$$x_i^{(j+1)} \sim \frac{\max(0, p_\theta(x|\boldsymbol{x}_{1:i-1}^{(j)}) - p_\theta(x|\boldsymbol{x}_{1:i-1}^{(j-1)}))}{\sum_x \max(0, p_\theta(x|\boldsymbol{x}_{1:i-1}^{(j)}) - p_\theta(x|\boldsymbol{x}_{1:i-1}^{(j-1)}))}. \tag{2}$$

Then, unlike the vanilla speculative sampling, we do not end this scan, but sample the tokens at the remaining indexes with the conditional probability calculated in this iteration. This sampling process is consistent with the original Jacobi iteration, and the specific process is as follows:

$$x_{i'}^{(j+1)} \sim p_\theta(x|\boldsymbol{x}_{1:i'-1}^{(j)}), \ \ \forall i' > i. \tag{3}$$

It can be proven that all the accepted and sampled tokens in the Jacobi iteration satisfy $x_i^{(j+1)} \sim p_\theta(x|\boldsymbol{x}_{1:i-1}^{(j)})$ (the proof is in the appendix). This conditional probability $p_\theta(x|\boldsymbol{x}_{1:i-1}^{(j)})$ is exactly the probability predicted by the parallel forward pass on the input draft tokens, and is passed to the next iteration together with the sampled tokens.

**Step 4**: we append the unaccepted tokens with newly initialized candidate tokens, forming a new Jacobi window with $W$ tokens, as the draft tokens for the next iteration. We use this fixed window size instead of the whole sequence to save the memory usage and accelerate the inference speed.

## 4.2 TOKEN INITIALIZATION WITH SPATIAL PRIOR

Vanilla Jacobi decoding methods sample the initial candidate tokens from a uniform distribution. However, 2D images exhibit unique characteristics of spatial locality, *i.e.*, spatially adjacent tokens tend to share similar semantics and textures. Leveraging these characteristics for token initialization may enable faster convergence. Considering that auto-regressive models generate image tokens in a raster scan order (from the top-left to the bottom-right in 2D space), we propose the following strategies for initializing new tokens: (a) repeating the previously generated left adjacent token; (b) repeating the previously generated above adjacent token; (c) resampling from the predicted probability from the left adjacent token; (d) resampling from the predicted probability from the above adjacent token. Experimental results demonstrate that these strategies provide greater acceleration than random initialization under certain scenarios.

## 5 EXPERIMENTS

### 5.1 IMPLEMENTATION DETAILS

We experiment with two recent and representative auto-regressive text-to-image generation models, Lumina-mGPT (Liu et al., 2024b) and Anole (Chern et al., 2024). For Lumina-mGPT (Liu et al.,

Table 1: The evaluation on the validation set of MSCOCO2017 with A100. `JD`: Jacobi decoding. `ISP`: initialization with spatial prior. `SJD`: Speculative Jacobi decoding.

| Configuration | | Average Latency (↓) | Acceleration (↑) | | FID (↓) | CLIP-Score (↑) |
|---|---|---|---|---|---|---|
| | | | Latency | Step | | |
| **A** | Lumina-mGPT (Liu et al., 2024b) | 87.23s | 1.00× | 1.00× | 30.76 | 31.29 |
| **B** | *w.* JD (Song et al., 2021) | 85.20s | 1.02× | 1.04× | 30.66 | 31.38 |
| **C** | *w.* **SJD** | 42.73s | 2.04× | 2.22× | 30.85 | 31.35 |
| **D** | *w.* **SJD (ISP)** | **42.49s** | **2.05×** | **2.23×** | 31.13 | 31.33 |
| **E** | Anole (Chern et al., 2024) | 48.96s | 1.00× | 1.00× | 28.87 | 30.59 |
| **F** | *w.* **SJD (ISP)** | **26.18s** | 1.87× | 1.97× | 29.14 | 30.61 |

Table 2: The evaluation on the validation set of Parti-prompt with RTX4090. `JD`: Jacobi decoding. `ISP`: initialization with spatial prior. `SJD`: Speculative Jacobi decoding.

| Configuration | | Average Latency (↓) | Acceleration (↑) | | CLIP-Score (↑) |
|---|---|---|---|---|---|
| | | | Latency | Step | |
| **A** | Lumina-mGPT (Liu et al., 2024b) | 100.69s | 1.00× | 1.00× | 32.13 |
| **B** | *w.* JD (Song et al., 2021) | 100.00s | 1.01× | 1.04× | 32.17 |
| **C** | *w.* **SJD** | 47.52s | 2.12× | 2.26× | 32.13 |
| **D** | *w.* **SJD (ISP)** | **47.35s** | **2.13×** | **2.28×** | 32.06 |
| **E** | Anole (Chern et al., 2024) | 48.24s | 1.00× | 1.00× | 30.46 |
| **F** | *w.* **SJD (ISP)** | **25.12s** | 1.92× | 2.11× | 30.48 |

2024b), by default, we use its 7B version to generate $768 \times 768$ images for evaluation, and we measure the sampling *randomness* by the value $K$ of its top-$K$ logit sampler. Following the basic setting of Lumina-mGPT, $K$ is set to 2000 and the classifier-free guidance weight is set to 3.0. Anole (Chern et al., 2024) is another 7B auto-regressive generation model finetuned from Chameleon (Team, 2024) that can generate $512 \times 512$ images.

**Metrics.** For visual quality, we use FID (Heusel et al., 2017) and CLIP-Score (Radford et al., 2021) as the metrics for evaluation. We use the *step compression ratio* (Fu et al., 2024b): $\mathcal{S} = \frac{\text{\# generated tokens}}{\text{\# decoding steps}}$ to show the theoretical acceleration ratio. For each benchmark, we report the average of the step compression ratio on all generated images. We also attach this ratio to each image sample in the qualitative comparison of our method with other approaches. Moreover, we also report the latency acceleration of the model forward passes on a single GPU for testing the actual speedup.

**Benchmark.** The parti-prompts (Yu et al., 2022) and the validation set of MS-COCO 2017 (Lin et al., 2014) are taken as the benchmarks of image generation. On parti-prompts, we use the CLIP-Score and the acceleration of latency and steps excluding FID for evaluation because this benchmark only provides prompts without ground-truth images.

## 5.2 QUANTITATIVE RESULTS

As shown in Tab. 1 and Tab. 2, our speculative Jacobi decoding accelerates the auto-regressive text-to-image generation nearly without sacrificing visual qualities. When comparing our SJD (**config C** and **D**) with the vanilla Jacobi decoding (**config B**) on Lumina-mGPT, we observe that our probabilistic method greatly accelerates the generation by more than $2\times$ while the Jacobi decoding cannot. Moreover, our method can provide a step compression of about $2\times$ for Anole. We observe that the token initialization with spatial priors has a marginal influence on the speed of general image generation. We further analyze the specific scenarios of this modification in our ablation studies.

## 5.3 QUALITATIVE RESULTS

As shown in Fig. 5, we present the images generated with different configurations. For comparison, we set the same random seed for each image sample. According to our observation, the visual qualities of the images generated by different methods are similar, illustrating that our method can keep the visual quality for multiple styles of images. More importantly, our speculative Jacobi decoding with or without our spatial initialization can greatly reduce the inference steps by more than $2\times$ for each case, and thus accelerate the inference process.

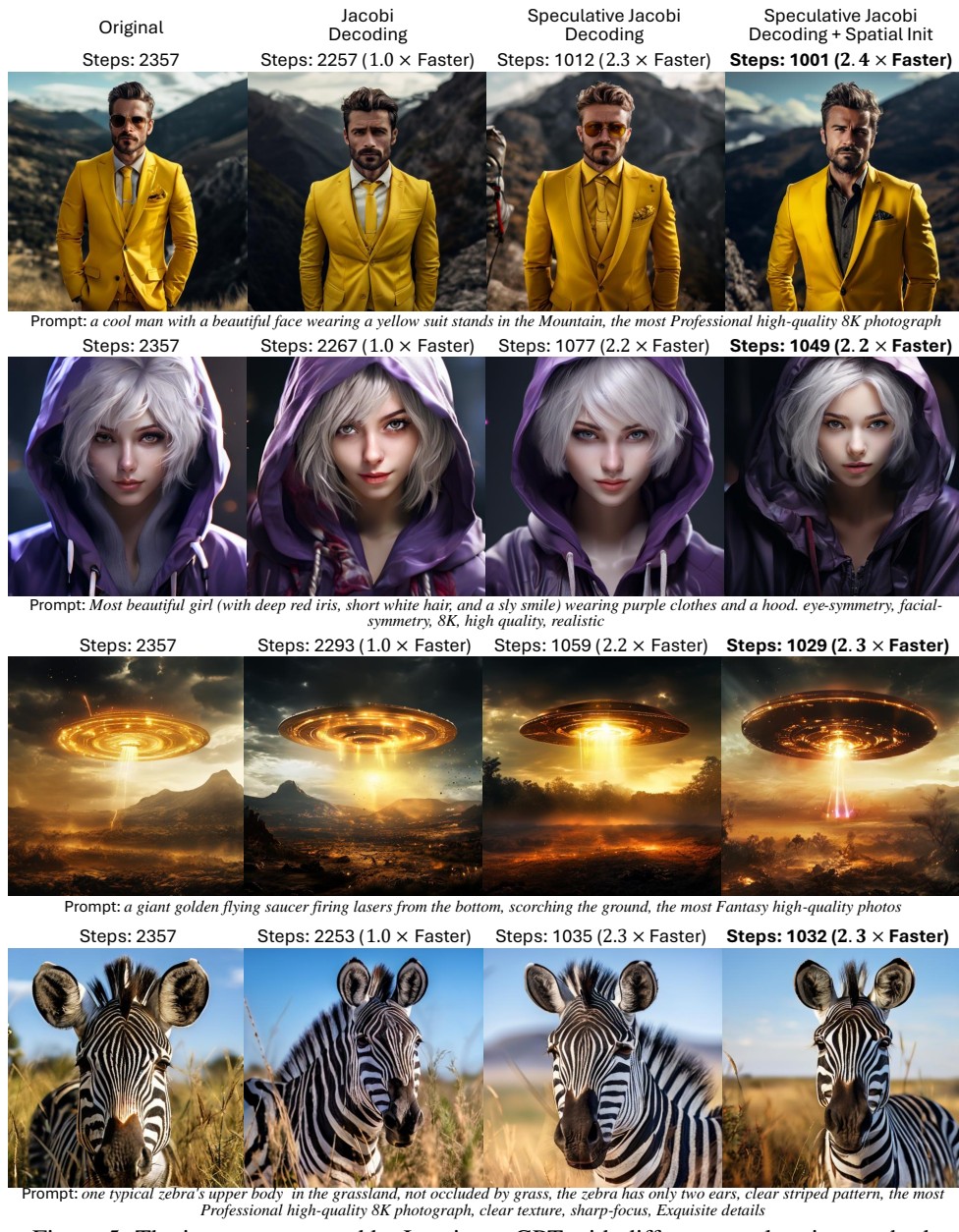

Figure 5: The images generated by Lumina-mGPT with different acceleration methods.

## 5.4 ABLATION STUDIES

We perform ablation studies on Lumina-mGPT 7B. Except for the experiments involving multiple resolutions, we use this model to generate $768 \times 768$ images for evaluation.

**The correlation between the sampling strategy in decoding and the acceleration ratio.** We compare the deterministic Jacobi decoding to our method under various randomness of the logit sampling. In Fig. 6, We show the correlation between their acceleration ratio and the randomness of logit sampling. According to this figure, our method is stable across multiple randomness and can achieve more than $2\times$ step compression ratio. On the contrary, the Jacobi decoding can only accelerate the greedy sampling (top-1 sampling), which is useless for image generation.

**The relationship between the image resolution and the acceleration ratio.** We employ the 7B Lumina-mGPT to generate images with the resolutions $512 \times 512$ (about 1,000 tokens), $768 \times 768$ (about 2,300 tokens), and $1024 \times 1024$ (about 4,100 tokens). We calculate the average step compression ratio for each resolution given the same set of text prompts. Then, we present these

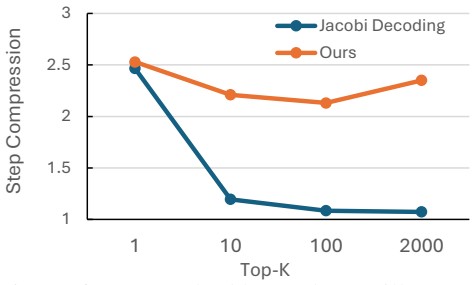

Figure 6: Our method beats the vanilla Jacobi decoding under various sampling randomness.

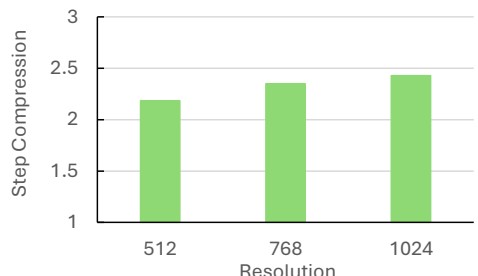

Figure 7: Higher image resolution can result in a slightly larger acceleration in our method.

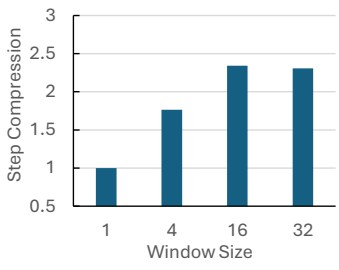

Figure 8: The acceleration ratio is the largest when the Jacobi window size is at least 16.

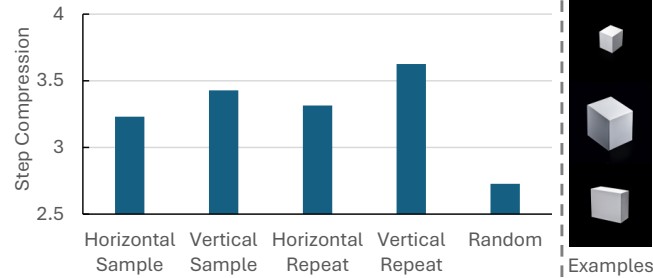

Figure 9: The token initialization strategy impacts the acceleration ratio of image generation that contains simple and repeat patterns (examples of generated images on the right side).

ratios in Fig. 7. The results demonstrate that our method is stable across multiple resolutions, *i.e.*, the acceleration on each resolution is larger than $2\times$. Moreover, with higher resolutions, the acceleration can be slightly better. For example, SJD achieves $2.43\times$ acceleration for $1024 \times 1024$ images.

**Studies on the window size of each iteration.** As mentioned in Sec. 4.1, we append newly initialized tokens onto the unaccepted tokens in each iteration, fixing the Jacobi window size. Accordingly, we perform the ablation studies on the size of the window. We report the acceleration ratios under various sequence lengths in Fig. 8. The results show that our acceleration ratio reaches almost the maximum when the number of input tokens is greater than or equal to 16 tokens.

**Studies on the initialization of candidate tokens.** The acceleration ratio of our speculative Jacobi decoding is also related to the application scenarios. For example, when generating images composed of many simple and repeating patterns, a token initialization correlated with the already sampled tokens can provide a more precise guess than the random initialization. As shown in Fig. 9, we adopt an extreme case, the textual prompt "2D logo of a pure white box in a pure black background", for evaluation. We run the accelerated forward passes ten times with different random seeds for each initialization. The results show that the average step compression with the spatial-prior-aware initialization is much greater than that using the random initialization. Also, Fig. 1 shows that generating `2D logo` requires fewer steps than generating images containing exquisite details.

## 6 CONCLUSION

This paper proposes a new training-free probabilistic parallel decoding algorithm, called **Speculative Jacobi Decoding (SJD)**, to accelerate auto-regressive text-to-image generation. The sampling-based decoding is critical for image generation models, which prevents naive Jacobi Decoding from being applied to accelerate auto-regressive text-to-image generation models. By introducing a probabilistic convergence criterion, our SJD allows the model to iteratively predict-then-sample multiple tokens in fewer steps than the token counts for auto-regressive text-to-image generation models with sampling-based decoding rather than greedy decoding. We also propose the spatial-aware token initialization to reduce the number of iterations under specific scenarios. We conduct experiments to verify the effectiveness of SJD on multiple auto-regressive text-to-image generation models, and it accelerates the models without sacrificing visual quality.

ACKNOWLEDGE

This work is supported by the National Nature Science Foundation of China (No. 62402406) and HKU IDS research Seed Fund. This work is partly supported by HKU Shanghai Intelligent Computing Research Center.

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

APPENDIX

## A  PROOFS

**Theorem 1** (The correctness of speculative Jacobi decoding) The token sampled in each speculative Jacobi iteration satisfies $p_\theta(x|\boldsymbol{x}_{1:i-1}^{(j)})$, where $x$ denotes a token, $j$ denotes the index of iteration, $i$ denotes the token index, and $\theta$ denotes the auto-regressive model parameters.

*Proof.* The main process of speculative Jacobi iteration is decomposed into two cases: (a) obtaining the token sampled in the previous iteration and then accepting it according to an acceptance probability; (b) rejecting the sampled token and resampling a new token according to a calibrated probability. Thus, like the proof of the vanilla speculative sampling (Leviathan et al., 2023), to prove the correctness of speculative Jacobi decoding, we verify that the conditional probability of a token sampled following the above two cases, alongside the manually designed acceptance and resampling probability, remains $p_\theta(x|\boldsymbol{x}_{1:i-1}^{(j)})$.

For simplicity, by default, we omit the token index $i$ and denote the token category of $\boldsymbol{x}_i^{(j)}$ as $x$. We denote the condition of token $\boldsymbol{x}_i^{(j)}$ at the $j$-th Jacobi iteration (*i.e.*, the tokens $\boldsymbol{x}_{1:i-1}^{(j)}$ and model weights $\theta$) to $\mathcal{J}_j$. Thus, the condition of the $(j-1)$-th Jacobi iteration is denoted as $\mathcal{J}_{j-1}$. Thus, we can denote the probability $p_\theta(x|\boldsymbol{x}_{1:i-1}^{(j)})$ as $p(x|\mathcal{J}_j)$, and denote $p_\theta(x|\boldsymbol{x}_{1:i-1}^{(j-1)})$ as $p(x|\mathcal{J}_{j-1})$. We use a random boolean variable $r$ to represent the acceptance. With these notations, the proof is as follows:

First, the acceptance probability on the token category $x$ is manually set as follows:

$$p(r \text{ is true}|x, \mathcal{J}_j, \mathcal{J}_{j-1}) = \min\{1, \frac{p(x|\mathcal{J}_j)}{p(x|\mathcal{J}_{j-1})}\}, \tag{4}$$

and the calibrated resampling probability subsequent to the rejection is set as follows:

$$p(x|r \text{ is false}, \mathcal{J}_j, \mathcal{J}_{j-1}) = \frac{\max\{0, p(x|\mathcal{J}_j) - p(x|\mathcal{J}_{j-1})\}}{\sum_{x'} \max\{0, p(x'|\mathcal{J}_j) - p(x'|\mathcal{J}_{j-1})\}}. \tag{5}$$

Next, we make an assumption that $\mathcal{J}_j$ and $x$ are conditionally independent given $\mathcal{J}_{j-1}$:

$$p(\mathcal{J}_j|x, \mathcal{J}_{j-1}) = p(\mathcal{J}_j|\mathcal{J}_{j-1}) \tag{6}$$

This assumption is reasonable due to the properties of the Jacobi iteration and the auto-regressive paradigm, *i.e.*, with the observation of the sequence $\boldsymbol{x}_{1:i-1}^{(j-1)}$, one of the tokens in $\boldsymbol{x}_{1:i-1}^{(j)}$ (denoted as $\boldsymbol{x}_k^{(j)}$) can be determined by $\boldsymbol{x}_k^{(j)} = f(\boldsymbol{x}_{1:k-1}^{(j-1)}, \theta)$ $(k < i)$ where the function $f$ indicates the prediction-then-sampling of auto-regressive models, so the variable $\boldsymbol{x}_i^{(j)}$ is redundant as one of the conditions in the probability $p(\mathcal{J}_j|x, \mathcal{J}_{j-1})$. Thus, Equ. (6) is reasonable.

Then, with Bayes rule, Equ. (6) has the following equivalence:

$$p(\mathcal{J}_j|x, \mathcal{J}_{j-1}) = p(\mathcal{J}_j|\mathcal{J}_{j-1}) \iff p(x|\mathcal{J}_j, \mathcal{J}_{j-1}) = p(x|\mathcal{J}_{j-1}) \tag{7}$$

Hence, according to Equ. (4) and Equ. (7), the probability that a token category $x$ is sampled in the previous iteration and subsequently accepted can be computed as:

$$\begin{aligned}
p(r \text{ is true}, x|\mathcal{J}_j, \mathcal{J}_{j-1}) &= p(x|\mathcal{J}_j, \mathcal{J}_{j-1}) \cdot p(r \text{ is true}|x, \mathcal{J}_j, \mathcal{J}_{j-1}) \\
&= p(x|\mathcal{J}_{j-1}) \cdot \min\{1, \frac{p(x|\mathcal{J}_j)}{p(x|\mathcal{J}_{j-1})}\} \\
&= \min\{p(x|\mathcal{J}_j), p(x|\mathcal{J}_{j-1})\}
\end{aligned} \tag{8}$$

Steps: 8193 → 3515 (**2.3 × Faster**)    Steps: 8193 → 3581 (**2.3 × Faster**)    Steps: 8193 → 3472 (**2.4 × Faster**)

Figure 10: The images generated by Emu3 (BAAI, 2024) with our acceleration method.

With Equ. (8), we can calculate the probability of rejection with the law of total probability on the token categories:

$$
\begin{aligned}
p(r \text{ is false}|\mathcal{J}_j, \mathcal{J}_{j-1}) &= 1 - p(r \text{ is true}|\mathcal{J}_j, \mathcal{J}_{j-1}) \\
&= 1 - \sum_{x'} p(r \text{ is true}, x'|\mathcal{J}_j, \mathcal{J}_{j-1}) \\
&= \sum_{x'} p(x'|\mathcal{J}_j) - \min\{p(x'|\mathcal{J}_j), p(x'|\mathcal{J}_{j-1})\} \\
&= \sum_{x'} \max\{0, p(x'|\mathcal{J}_j) - p(x'|\mathcal{J}_{j-1})\}.
\end{aligned}
\tag{9}
$$

Then, with Equ. (5) and Equ. (9), we get the following equation:

$$
\begin{aligned}
&p(x|r \text{ is false}, \mathcal{J}_j, \mathcal{J}_{j-1}) \cdot p(r \text{ is false}|\mathcal{J}_j, \mathcal{J}_{j-1}) \\
&= \frac{\max\{0, p(x|\mathcal{J}_j) - p(x|\mathcal{J}_{j-1})\}}{\sum_{x'} \max\{0, p(x'|\mathcal{J}_j) - p(x'|\mathcal{J}_{j-1})\}} \cdot \sum_{x'} \max\{0, p(x'|\mathcal{J}_j) - p(x'|\mathcal{J}_{j-1})\} \\
&= \max\{0, p(x|\mathcal{J}_j) - p(x|\mathcal{J}_{j-1})\}.
\end{aligned}
\tag{10}
$$

Since

$$
\forall a \in \mathbb{R}, b \in \mathbb{R}, \ a = \min\{a, b\} + \max\{0, a - b\},
\tag{11}
$$

we can decompose $p(x|\mathcal{J}_j)$ as follows:

$$
p(x|\mathcal{J}_j) = \min\{p(x|\mathcal{J}_j), p(x|\mathcal{J}_{j-1})\} + \max\{0, p(x|\mathcal{J}_j) - p(x|\mathcal{J}_{j-1})\}.
\tag{12}
$$

With Equ. (8), Equ. (10) and Equ. (12), we can compute:

$$
\begin{aligned}
p(x|\mathcal{J}_j) &= \min\{p(x|\mathcal{J}_j), p(x|\mathcal{J}_{j-1})\} + \max\{0, p(x|\mathcal{J}_j) - p(x|\mathcal{J}_{j-1})\} \\
&= p(x|\mathcal{J}_{j-1}) \cdot p(r \text{ is true}|x, \mathcal{J}_j, \mathcal{J}_{j-1}) \\
&\quad + p(r \text{ is false}|\mathcal{J}_j, \mathcal{J}_{j-1}) \cdot p(x|r \text{ is false}, \mathcal{J}_j, \mathcal{J}_{j-1}).
\end{aligned}
\tag{13}
$$

According to Equ. (13), the conditional distribution $p(x|\mathcal{J}_j)$ can exactly represent (a) obtaining the token sampled in the previous iteration and then accepting it according to an acceptance probability; (b) rejecting the sampled token and resampling a new token according to a calibrated probability. In conclusion, the token sampled in each speculative Jacobi iteration satisfies $p_\theta(x|\boldsymbol{x}_{1:i-1}^{(j)})$.

## B    MORE QUALITATIVE RESULTS

In Fig. 11, we showcase more generated images with Lumina-mGPT accelerated by our method. These results illustrate that our method functions well on the image contents including humans, animals, and landscapes. Recently, a new powerful auto-regressive model, Emu3 (BAAI, 2024), has been released. We also explore our method on Emu3 for text-to-image generation, and we find it

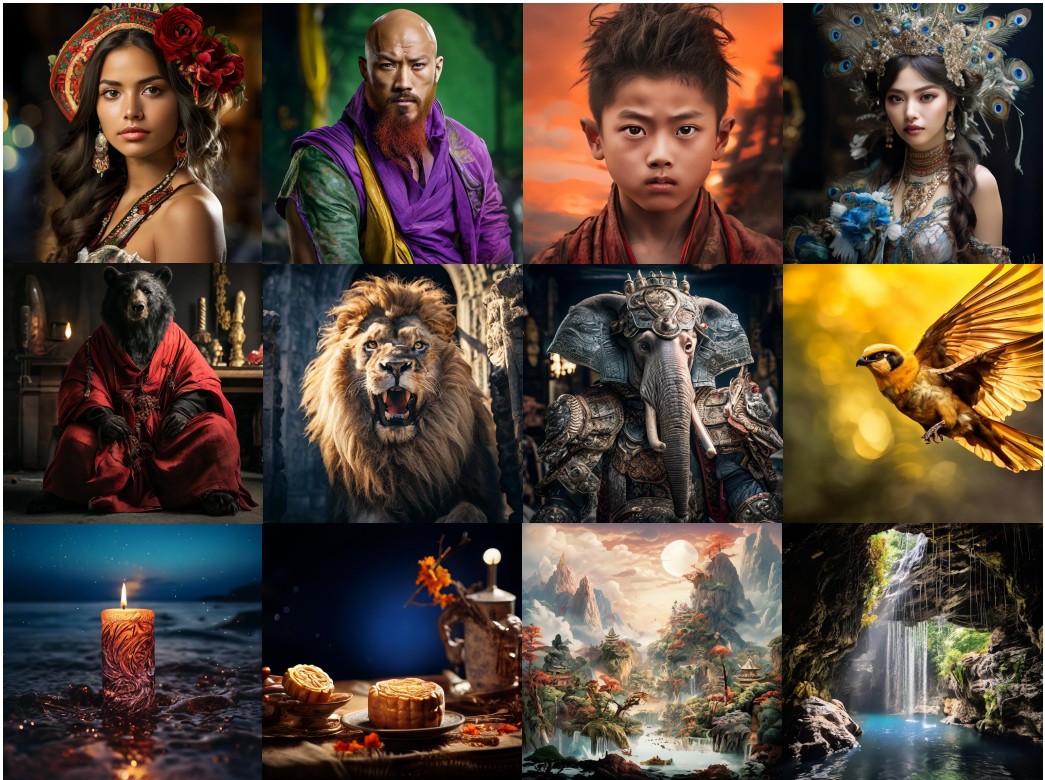

Figure 11: The images generated by Lumina-mGPT (Liu et al., 2024b) with our acceleration method.

still leads to great step compression, shown in Fig. 10. We leave the quantitative results of Emu3 for future work.

We have included additional qualitative results for Lumina-mGPT and Anole in the supplementary material of the revised paper, specifically in Fig. 15 and Fig. 16, and we report both the steps and latency. According to the reported latency and step compression in these figures, our SJD outperforms other decoding methods while maintaining visual quality. Furthermore, spatial token initialization can further enhance the acceleration of our SJD. Additionally, we observe that Anole exhibits significantly higher image diversity compared to Lumina-mGPT. Despite the fixed random seed, it remains challenging for Anole to generate similar images due to the differences among the decoding methods.

## C    INFERENCE LATENCY

In addition to reporting the step compression ratio, we also report the practical latency of SJD on servers. We set the batch size as 1 for testing, and report the latency of the accelerated Lumina-mGPT 7B excluding the pre- and post-processing operations. For $768 \times 768$ image generation (the number of generated tokens is at least 2357), we perform the experiments on one RTX 4090 GPU. For $1024 \times 1024$ image generation (the number of generated tokens is at least 4165), we perform the experiments on one A100 GPU. In these settings, the latency of Lumina-mGPT with and without our method is presented in Fig. 12. Our method significantly accelerates the auto-regressive image generation.

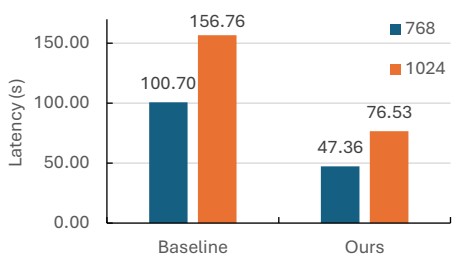

Figure 12: The latency of Lumina-mGPT on generating $768 \times 768$ and $1024 \times 1024$ images without or with our method.

Table 3: The evaluation of LlamaGen (Sun et al., 2024a) with or without our method on MSCOCO2017 (Lin et al., 2014) and Parti-prompt (Yu et al., 2022).

| Dataset | Configuration | Acceleration (↑) | | FID (↓) | CLIP-Score (↑) |
| --- | --- | --- | --- | --- | --- |
| | | Latency | Step | | |
| COCO | LlamaGen-stage1 | 1.00× | 1.00× | 28.54 | 30.87 |
| | LlamaGen-stage1 + Ours | 1.56× | 1.63× | 29.00 | 30.82 |
| | LlamaGen-stage2 | 1.00× | 1.00× | 56.21 | 28.26 |
| | LlamaGen-stage2 + Ours | 1.54× | 1.63× | 57.02 | 28.33 |
| Parti | LlamaGen-stage1 | 1.00× | 1.00× | - | 30.22 |
| | LlamaGen-stage1 + Ours | 1.57× | 1.73× | - | 30.29 |
| | LlamaGen-stage2 | 1.00× | 1.00× | - | 28.14 |
| | LlamaGen-stage2 + Ours | 1.62× | 1.69× | - | 28.16 |

## D MORE QUANTITATIVE RESULTS

**More Results**. We further compare SJD to other decoding methods on Anole (Chern et al., 2024). As shown in Tab. 4 and Tab. 5, consistent with the results on Lumina-mGPT, SJD with spatial token initialization can create larger acceleration ratios than other decoding methods on Anole, and the cost of visual quality is small. In addition to Anole (Chern et al., 2024) and Lumina-mGPT (Liu et al., 2024b), we evaluate our method with the text-to-image LlamaGen (Sun et al., 2024a). This model adopts a two-stage training strategy: (a) stage1: LlamaGen is first trained on a subset of LAION-COCO (LAION, 2022) (50M $256 \times 256$ images); (b) stage2: it is then fine-tuned on 10M high aesthetic quality internal data with a resolution of $512 \times 512$. In Tab. 3, we evaluate our method with the two versions of LlamaGen. The results show that our method can still accelerate this model without sacrificing the visual quality. However, in comparison to the experiments conducted on Lumina-mGPT and Anole, the acceleration ratios on LlamaGen are lower. We hypothesize that this discrepancy is attributed to the model size, as some existing works for multi-token prediction demonstrate that the model size has a great influence on the effectiveness of acceleration (Gloeckle et al., 2024). We leave this investigation to future work.

**More results about visual quality**. We take the CLIP-Score and the human preference score (HPSv2) (Wu et al., 2023) as the metrics for evaluating the visual quality for our ablation studies (the step compression ratios are reported in Sec. 5.4). We present the results in Tab. 7, Tab. 8, Tab. 9, and Tab. 10. From Tab. 7, given any $K$ values in the top-$K$ sampling strategies, we can observe that the human preferences are also not much different among the original auto-regressive decoding, the original Jacobi decoding, and our SJD.

**Perplexity**. We also compare the perplexities between SJD and other decoding methods on Lumina-mGPT, as detailed in Tab. 6. Since the perplexities are influenced by the sampling strategies (Hu et al., 2023), we report the perplexities under various $K$ values. Given an identical $K$ value, the perplexities between our method and other decoding methods are close. Furthermore, we note that $K = 2000$ results in a perplexity higher than that of large language models (Gu & Dao, 2023; Yang et al., 2024b) on language processing tasks. Despite this high value, the text-to-image auto-regressive model can still generate high-quality images. This indicates that image generation can tolerate a wide range of image tokens.

**Statistics of model outputs**. We compute the statistics of the logarithm of the token probability for both auto-regressive decoding and our method. The average and standard deviation of all image tokens are presented in Tab. 11. The results demonstrate that the image tokens accepted by our method exhibit similar statistics to those accepted by the original auto-regressive decoding. Consequently, our method generally does not mistakenly accept tokens with lower probabilities.

## E VISUALIZATION OF ACCELERATION IN 2D SPACE

We visualize the impact of multi-token prediction in a 2D space. As illustrated in Fig. 13, the color of each long strip area represents the length of accepted tokens from that area, with darker colors indicating longer sequences of accepted tokens, *i.e.*, higher acceleration. We observe that

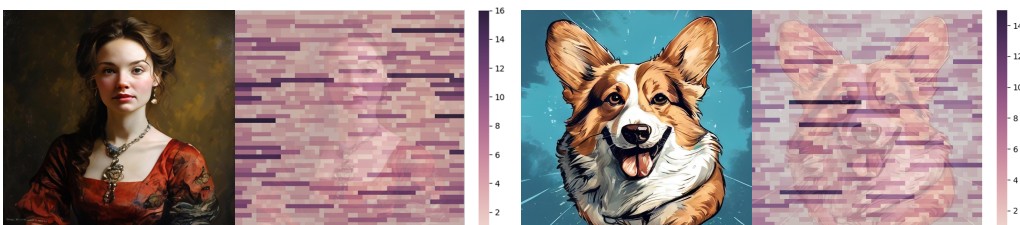

Figure 13: The visualization of the accelerated tokens on 2D space.

high acceleration tends to occur in the background, particularly on the left and right sides of images. Additionally, while some high acceleration is observed on foreground objects, it is relatively sparse in 2D space.

## F    Analysis on the Effectiveness of our method

This section analyzes the acceleration mechanism of our speculative Jacobi decoding in image generation. *We empirically find that this acceleration stems from the resampling of unaccepted tokens.* Specifically, some tokens are *continuously resampled* (*i.e.*, ***their positions within the entire sequence are reused for multiple forward passes***) according to Equ. (3) over iterations until they are accepted. For clarity and simplicity, we refer to this process of a token being continuously resampled by Equ. (3) (except the possible rejection resampling) as *refinement*, following the terminology in fixed-point iteration (Bai et al., 2019; 2022; Wang et al., 2023). Consequently, Equ. (3) is the main operation of every *refinement step*. In the following paragraphs, we explore the influences of this refinement.

**The acceleration originates from the refinement of unaccepted tokens.**    In our verification phase, there are *three treatments* for the tokens: *acceptance*, *rejection*, and *refinement*, corresponding to Equ. (1), Equ. (2), and Equ. (3), respectively. We empirically find that ***only the first two treatments*** are ***insufficient*** to support acceleration. We conduct the following experiment to demonstrate that our method makes it hard to achieve acceleration without refinement: *when we deactivate the refinement (i.e., using the newly initialized tokens to replace the unaccepted tokens as the draft tokens in the next iterations), we observe that the model requires over two thousand forward passes to generate images rather than one thousand forward passes. Although our token initializations with spatial prior (e.g., horizontal repeat) are slightly better than the random token initialization in replacing the unaccepted tokens, its performance is still much worse than directly refining the unaccepted tokens.* The examples of the generated images under such setting are shown in Fig. 14. This phenomenon illustrates that the acceleration of our method originates from refining unaccepted tokens.

## G    Qualitative Analysis of image randomness on our method

Like Fig. 2, we also examine the image randomness with both the auto-regressive decoding and our speculative Jacobi decoding. As shown in Fig. 17, first, we find that SJD does introduce some randomness into image generation (the random variable $r$ in Equ. (1)), so the images generated with auto-regressive decoding cannot exactly align those generated with SJD, even when the random seed is fixed. Therefore, in Fig. 17, given a column, two images with the same $K$ value cannot be exactly identical. However, in general, the diversity of the set of images is not influenced much. In Fig. 17, we present the images generated based on three textual prompts. Given the same prompt and $K$ value from top-$K$ sampling, the model with different decoding methods generates images with many similarities. For example, when $K = 2000$, for the first prompt "an apple of a strange color", the images in the identical columns show the apples with similar color patterns and styles. Also, for the third prompt "pumpkin on the table", the frequency of faces carved on the pumpkins is similar for these two decoding methods.

Moreover, the $K$ value in top-$K$ sampling still dominates the image randomness in terms of texture, color, and local structure details. With larger $K$, the image details about textures, colors, and local structures increase. Such image randomness still largely comes from the random token sampling.

Examples of Images Generated (**Without** *Refinement*; **With random** *initialization*)

| Latency: 107.35s | Latency: 107.49s | Latency: 107.96s | Latency: 107.64s | Latency: 107.80s |
| Steps: 2356 | Steps: 2356 | Steps: 2357 | Steps: 2355 | Steps: 2357 |

Examples of Images Generated (**Without** *Refinement*; **With horizontal repeat** *initialization*)

| Latency: 99.25s | Latency: 93.88s | Latency: 95.08s | Latency: 95.60s | Latency: 95.42s |
| Steps: 2273 | Steps: 2254 | Steps: 2285 | Steps: 2300 | Steps: 2291 |

Examples of Images Generated (**With** *Refinement*; **With horizontal repeat** *initialization only for New Tokens*)

| **Latency: 42.49s** | **Latency: 42.52s** | **Latency: 41.95s** | **Latency: 41.23s** | **Latency: 42.44s** |
| **Steps: 1032** | **Steps: 1049** | **Steps: 969** | **Steps: 947** | **Steps: 1029** |

Figure 14: Ablation studies on acceleration mechanism: examples of images generated by our SJD without or with refining unaccepted tokens. When the refinement defined by Equ. (3) is **NOT** applied (*i.e.*, using the newly initialized tokens to replace the unaccepted tokens as the draft tokens in the next iterations), there is almost no acceleration (though one of our token initializations with spatial prior, horizontal repeat, can slightly reduce the steps in these images). This illustrates that **refining** unaccepted tokens are essential to the acceleration mechanism in SJD.

## H   ANALYSIS ON FAILURE CASES

As shown in Fig. 18, when generating the images with exquisite details, although auto-regressive decoding can produce artifacts, SJD seems to generate continuous tokens that cause the artifacts, as highlighted by the red boxes in this figure. The pre-trained auto-regressive model is not sufficiently robust to handle such complex images. Consequently, it may mistakenly accept a sequence of draft tokens that contain artifacts.

## I   LIMITATION AND FUTURE WORK

Since our speculative Jacobi decoding is training-free, the accelerated model itself is still not specialized for multi-token prediction. Therefore, the acceleration ratio has the potential to be further improved. In the future, we believe that fine-tuning the auto-regressive models for fast image generation is a promising direction. Also, acceleration is important for long-sequence generation, like video generation. Since videos contain more redundancy than images, the initialization of candidate tokens should be carefully designed if applying our speculative Jacobi decoding to video generation.

Table 4: The evaluation of Anole on the validation set of MSCOCO2017. `JD`: Jacobi decoding. `ISP`: initialization with spatial prior. `SJD`: Speculative Jacobi decoding.

| Configuration | Average Latency (↓) | Acceleration (↑) | | FID (↓) | CLIP-Score (↑) |
|---|---|---|---|---|---|
| | | Latency | Step | | |
| **A**   Anole (Chern et al., 2024) | 48.96s | 1.00× | 1.00× | 28.87 | 30.59 |
| **B**   *w.* JD (Song et al., 2021) | 47.60s | 1.03× | 1.06× | 29.34 | 30.64 |
| **C**   *w.* **SJD** | 27.08s | 1.81× | 1.94× | 29.04 | 30.54 |
| **D**   *w.* **SJD (ISP)** | **26.18s** | **1.87×** | **1.97×** | 29.14 | 30.61 |

Table 5: The evaluation of Anole on the validation set of Parti-prompt. `JD`: Jacobi decoding. `ISP`: initialization with spatial prior. `SJD`: Speculative Jacobi decoding.

| Configuration | Average Latency (↓) | Acceleration (↑) | | CLIP-Score (↑) |
|---|---|---|---|---|
| | | Latency | Step | |
| **A**   Anole (Chern et al., 2024) | 48.24s | 1.00× | 1.00× | 30.46 |
| **B**   *w.* JD (Song et al., 2021) | 44.65s | 1.08× | 1.14× | 30.57 |
| **C**   *w.* **SJD** | 26.77s | 1.80× | 2.00× | 30.55 |
| **D**   *w.* **SJD (ISP)** | **25.12s** | **1.92×** | **2.11×** | 30.48 |

## J   BROADER IMPACTS

Image generation offers extensive utility in helping users, designers, and artists produce fantastic content. Nonetheless, these models could be exploited to create deceptive content. Thus, it is crucial for the users including researchers and developers to acknowledge the potential negative social impact of image generation models.

Table 6: The comparison of perplexity on Lumina-mGPT.

| Configuration | Perplexity with Top-$K$ sampling | | |
| --- | --- | --- | --- |
| | $K = 10$ | $K = 100$ | $K = 2000$ |
| **A** Lumina-mGPT (Liu et al., 2024b) | 7.31 | 43.37 | 204.06 |
| **B** *w.* JD (Song et al., 2021) | 7.20 | 43.85 | 197.64 |
| **C** *w.* **SJD** | 7.34 | 43.87 | 217.96 |
| **D** *w.* **SJD** (**ISP**) | 7.26 | 44.03 | 199.70 |

Table 7: CLIP-Score of various decoding methods on Lumina-mGPT with different top-$K$ values. The image qualities for Jacobi Decoding and our method correspond to Fig. 6. The image qualities for Auto-regression are only for the comparison in this table. Note that the image quality score with greedy sampling is extremely poor, as this setting leads to meaningless images for a lot of prompts (analyzed in Fig. 2).

| Decoding Methods | Sampling | CLIP-Score | HPSv2 |
| --- | --- | --- | --- |
| Auto-regression | Top-1 Sampling | 26.40 | 0.1976 |
| Auto-regression | Top-10 Sampling | 32.83 | 0.2950 |
| Auto-regression | Top-100 Sampling | 32.41 | 0.3020 |
| Auto-regression | Top-2000 Sampling | 32.00 | 0.2965 |
| Jacobi Decoding | Top-1 Sampling | 26.34 | 0.1413 |
| Jacobi Decoding | Top-10 Sampling | 32.75 | 0.2960 |
| Jacobi Decoding | Top-100 Sampling | 32.46 | 0.3089 |
| Jacobi Decoding | Top-2000 Sampling | 31.68 | 0.3103 |
| Ours | Top-1 Sampling | 26.16 | 0.1695 |
| Ours | Top-10 Sampling | 32.27 | 0.2942 |
| Ours | Top-100 Sampling | 32.65 | 0.2977 |
| Ours | Top-2000 Sampling | 31.83 | 0.3020 |

Table 8: CLIP-Scores on Lumina-mGPT with various resolutions. The image qualities of our method under different settings correspond to Fig. 7. The image qualities for Auto-regression are only for the comparison in this table.

| Decoding Methods | Resolutions | CLIP-Score | HPSv2 |
| --- | --- | --- | --- |
| Auto-regression | 512 | 29.49 | 0.2503 |
| Auto-regression | 768 | 32.00 | 0.2965 |
| Auto-regression | 1024 | 31.41 | 0.2961 |
| Ours | 512 | 29.69 | 0.2558 |
| Ours | 768 | 31.83 | 0.3020 |
| Ours | 1024 | 31.11 | 0.2935 |

Table 9: CLIP-Score of our method on Lumina-mGPT with various Jacobi window sizes. The image qualities correspond to Fig. 8.

| Window Size | CLIP-Score | HPSv2 |
| --- | --- | --- |
| 1 | 32.00 | 0.2965 |
| 4 | 31.91 | 0.3046 |
| 16 | 31.83 | 0.3020 |
| 32 | 31.55 | 0.3045 |

Table 10: CLIP-Score of our method on Lumina-mGPT with various token initialization when generating images with simple patterns. The image qualities correspond to Fig. 9.

| Token Initialization | CLIP-Score | HPSv2 |
| --- | --- | --- |
| Horizontal Sample | 31.52 | 0.2567 |
| Vertical Sample | 30.91 | 0.2622 |
| Horizontal Repeat | 31.17 | 0.2616 |
| Vertical Repeat | 31.15 | 0.2651 |
| Random | 31.37 | 0.2681 |

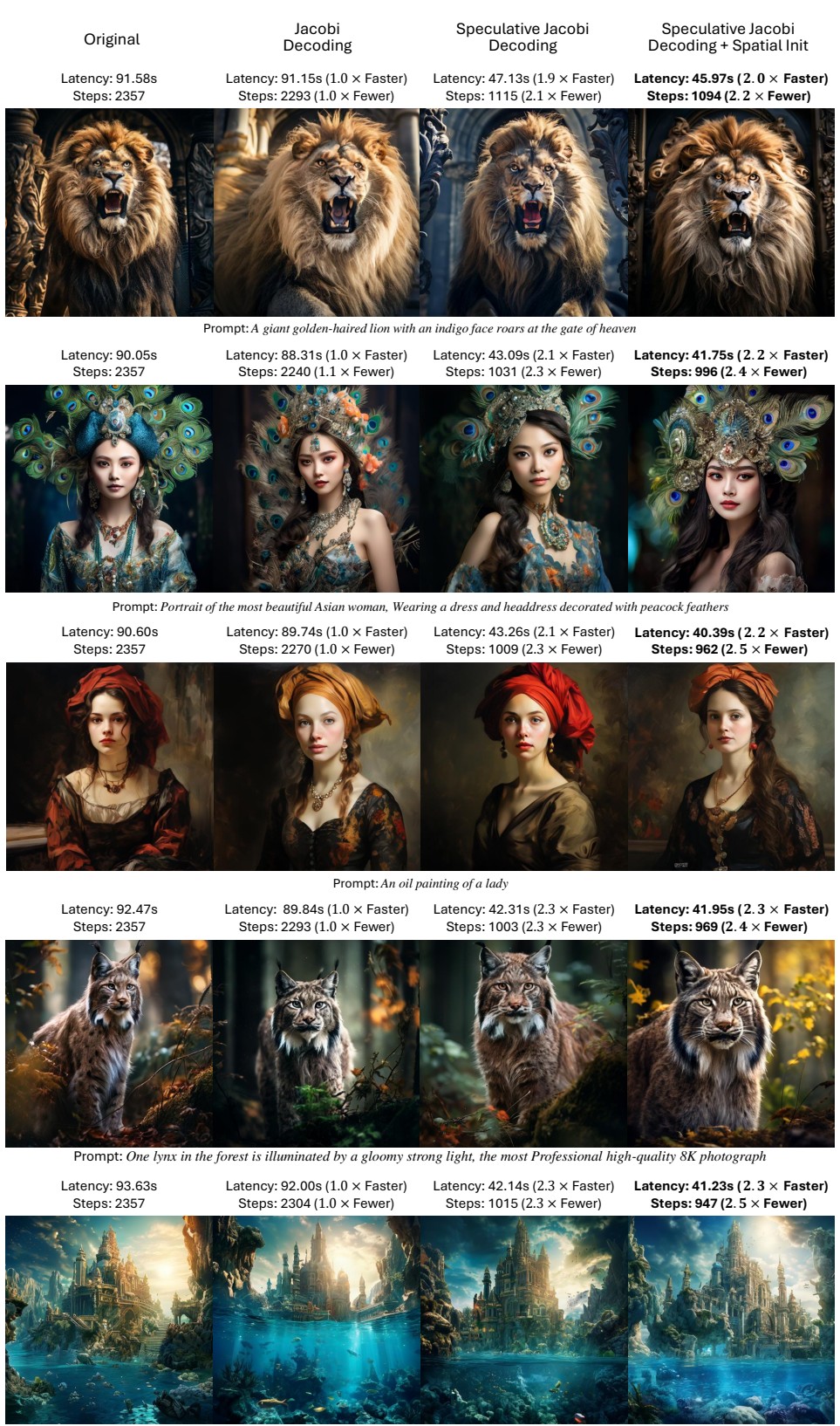

Figure 15: The qualitative comparison of different decoding methods on Lumina-mGPT (Liu et al., 2024b).

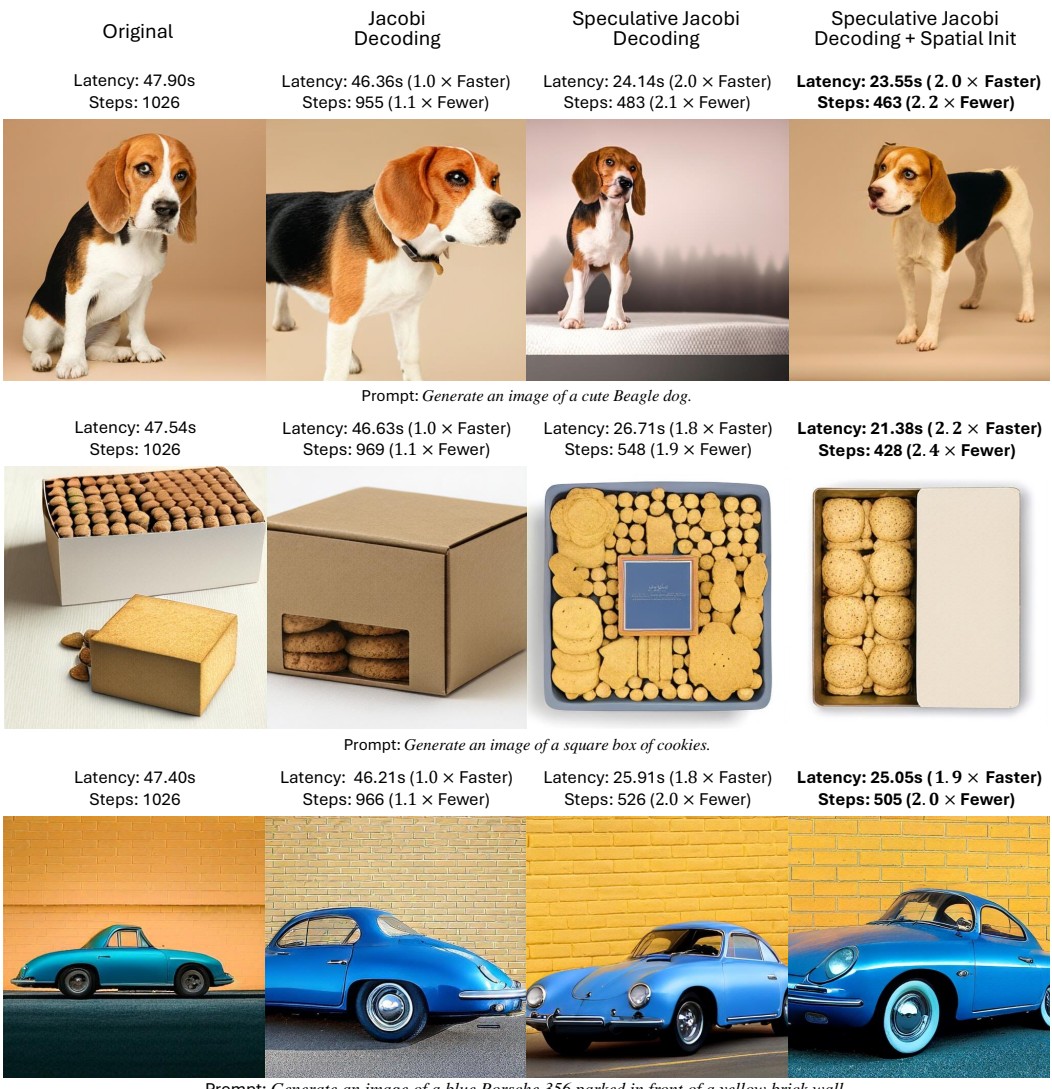

Figure 16: The qualitative comparison of different decoding methods on Anole (Chern et al., 2024). Considering the high image diversity of Anole, although the random seed is fixed, it is still hard for Anole to generate similar images with different decoding methods.

Table 11: The comparison of token statistics on Lumina-mGPT.

| Decoding Methods | Logarithm of Token Probability | |
| --- | --- | --- |
| | Average | Standard Deviation |
| Auto-regression | -4.8950 | 2.3457 |
| Ours | -4.9007 | 2.3275 |

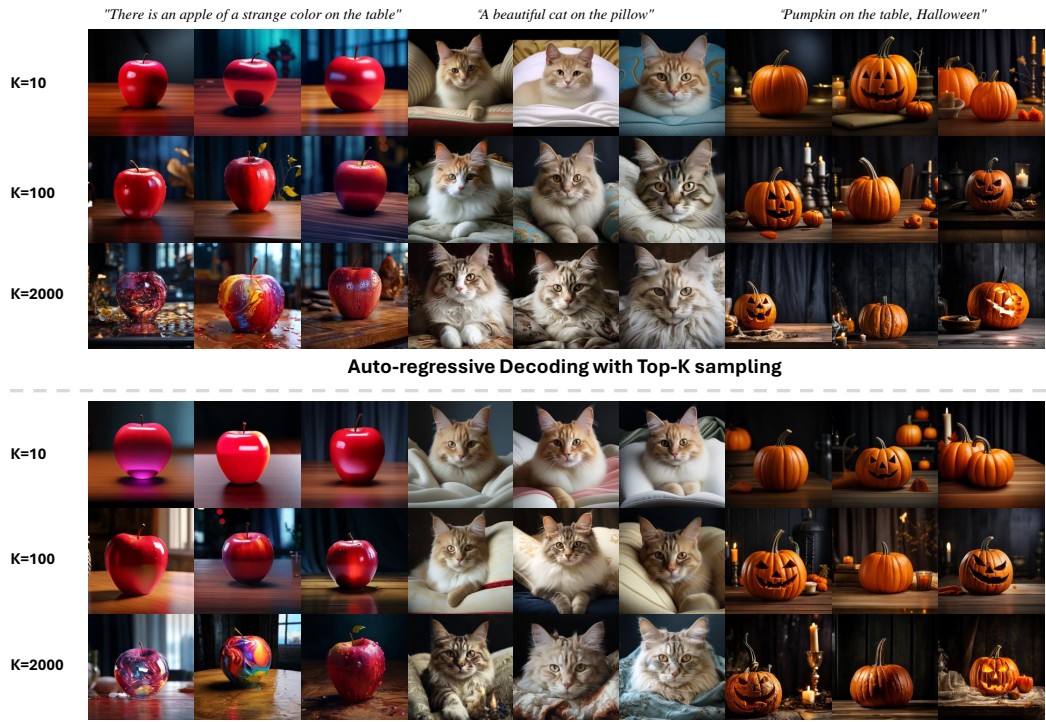

Figure 17: Comparing our method to the original auto-regressive decoding on the image randomness. First, considering the random variable in SJD, given a column, two images with the same $K$ value cannot be exactly identical. Second, changing the decoding method from auto-regression to SJD has little influence on the image diversity for each prompt (*e.g.*, given $K = 2000$ for each decoding method, the color patterns and styles of the generated apples are similar, and the frequency of the carved faces on pumpkins is also similar). Third, the top-$K$ sampling still dominates the image randomness about texture, color, and local structure details. The images in each column share a single random seed.

Prompt: *Image of a bustling downtown street in Tokyo at night, with neon signs, crowded sidewalks, and tall skyscrapers.*

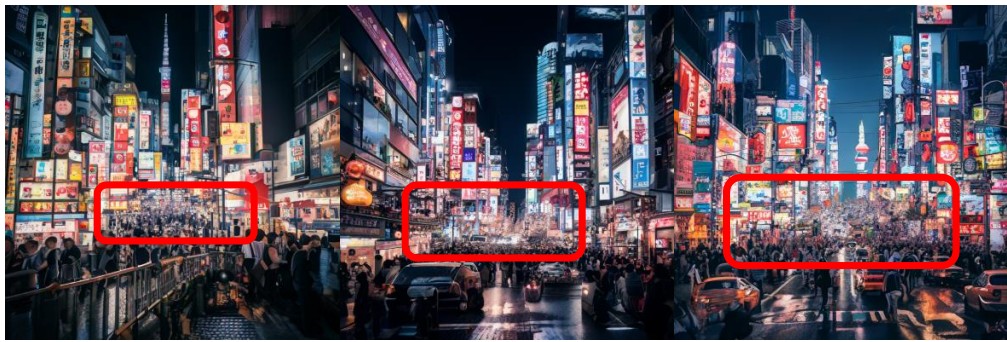

**Speculative Jacobi Decoding**

Figure 18: **Failure Cases**. In complex image scenarios, our method generates some continuous tokens that result in artifacts, as highlighted by the red boxes. The pre-trained model inaccurately accepts a large sequence of the tokens that cause the artifacts.

