# OpenReview forum: "Accelerating Auto-regressive Text-to-Image Generation with Training-free Speculative Jacobi Decoding"
_ICLR.cc/2025/Conference — ICLR 2025 Poster_

### Official Review · Reviewer_EsPc · 2024-10-27

**Soundness:** 2
**Presentation:** 2
**Contribution:** 2
**Rating:** 5
**Confidence:** 2

**Summary:**

SJD iteratively decodes multiple tokens in parallel within a sliding window. The model computes the conditional probability for a sequence of draft tokens in a single forward pass. A probabilistic criterion then determines which tokens to accept, adding them to the fixed pre-filling sequence. Unaccepted tokens are resampled for the next iteration. The spatial locality-aware token initialization strategy leverages the spatial relationships in images for initializing new tokens. SJD effectively accelerates auto-regressive text-to-image generation models, achieving approximately 2× speedup for Lumina-mGPT and Anole without significant visual quality degradation.

**Strengths:**

This paper introduces reject sampling to accelerate auto-regressive text-to-image generation.
SJD enables the model to predict multiple tokens at each step, accepting them based on a probabilistic criterion, reducing the number of inference steps compared to conventional next-token-prediction.
The paper proposes a spatial locality-aware token initialization strategy to further enhance acceleration in specific scenarios.
Experiments on multiple auto-regressive text-to-image generation models demonstrate SJD's effectiveness in accelerating inference without compromising visual quality.

**Weaknesses:**

1) SJD relax the greedy acceptence criteron of next token by reject sampling. Such acceleration will sample results with smaller probabilities and JD is also able to update multiple tokens at a time.  And relaxation of acceptence criteron lead to suboptimal results. Moreover, this paper is infact very simple with little experiments. It's more like a trick rathor than a paper.
2) The results that JD only accelerate t2i models by a rate of 1.04 is not convicing, which is much worse than results provided by JD and other papers.
3) The evaluation details are not provided. As far as I'm concerned, SJD may leads The nolvelty seems incremental. to suboptimal image quality and better image diversity. Hence, the FID score would be better if the image quality decrease a little and the image diversity increase.
4) There are too little models tested. It's better to apply SJD to more large models and small models.
Not that, I'm not familar to large model accelaration, I will consider other reviews' comments.

**Questions:**

Q1: Could the authors provide the conparison of perplex between SJD and other decoding methods?


Q2: I'm curious about how many image samples are used to compute the FID scores. Could the authors provide more evaluation details?
I doubt that the authors have use some tricks in evalution.  For example, the image quantity maybe different to compared method
Q3: Could the author provide more  results on small  models?
Not everyone has access to large models, it's more convicing to conduct experiments on small models

---

> ### Author Response · Authors · 2024-11-20
>
> We are grateful to the reviewer for the review of our submission. Below, we provide detailed responses to the questions and concerns raised.
>
> 1. **SJD relaxes the greedy acceptance criterion of JD, which might lead to suboptimal performance; About the contribution and novelty of our paper.**
>
>     This work is not a simple trick or naive improvement over Jacobi Decoding. Instead, we conduct a thorough analysis and provide deep insights into a problem that has been previously overlooked, proposing an effective solution to this problem. We begin by analyzing an intriguing phenomenon: Jacobi Decoding does not effectively accelerate auto-regressive text-to-image generation models. Through further analysis, we identified the key problem: the deterministic criterion of Jacobi Decoding is only effective with greedy decoding and is incompatible with the random sampling (e.g., top-$K$ sampling) used in text-to-image models. This random sampling is essential for generating diverse and high-quality images with sufficient details (as depicted in Figure 2). To address this problem, we introduce the probabilistic acceptance criterion into Jacobi Decoding. Our SJD does not lead to suboptimal performance compared to JD (Please refer to the table of human preference score below, where the scores on the decoding methods are similar), but rather it solves the problem that JD cannot accelerate auto-regressive text-to-image generation models under the common random sampling settings.
>     We would also like to emphasize that our SJD is a **simple yet effective** method for accelerating auto-regressive text-to-image generation. This **training-free** acceleration algorithm has been deployed on various auto-regressive text-to-image generation models, including **Lumina-mGPT, Anole, LlamaGen, and Emu3**, achieving notable acceleration ratios with minimal impact on image quality. To thoroughly discuss and analyze the acceleration and visual quality preservation of SJD across different models and settings, we have conducted extensive experiments to verify the effectiveness and generalization ability of our paper, encompassing 10 tables and 13 figures in the revised version of this paper.
>
>     | Decoding Methods  |  HPSv2 |
>     |-------------|-------------|
>     | Auto-regression  | 0.2965 |
>     | Jacobi Decoding  | 0.3103 |
>     | Ours  | 0.3020 |
>
> 2. **About the performance of the original Jacobi Decoding.**
>
>     Our experiments have demonstrated that the original Jacobi Decoding is **NOT** suitable for the current auto-regressive text-to-image generation. The original Jacobi Decoding paper demonstrated its success on MNIST and CIFAR-10 class-conditioned image generation, but these datasets are much smaller, and the class-conditioned image generation on these datasets requires far less image diversity and details compared to current text-to-image generation. On these small datasets, models can use greedy decoding to generate images, so Jacobi Decoding works well. However, the current text-to-image generation task requires higher diversity and more details in the generated images, and the current auto-regressive models exhibit different behaviors from the previous class-conditional auto-regressive models that Jacobi Decoding experimented on. Current auto-regressive models require top-$K$ sampling rather than greedy decoding, but Jacobi Decoding fails to perform well on top-$K$ sampling.
>     As shown in Figure 6, our ablation study demonstrates that the acceleration ratio of the original Jacobi Decoding rapidly decreases from about 2.5 to 1.0 as the $K$ values in top-$K$ sampling increase from 1 to 2000. This phenomenon illustrates that Jacobi Decoding is effective only with greedy sampling, which is not practical for current auto-regressive text-to-image generation (as shown in Figure 2, we explained why greedy sampling is unsuitable for current auto-regressive text-to-image generation).
>     We are confident of our implementation and experimental results, and we will open-source our code for the community.

---

> ### Author Response · Authors · 2024-11-20
>
> 3. **About the evaluation details about image quality**
>
>     To obtain the FID score on the MSCOCO 2017 validation set (comprising 5,000 ground-truth images), we first run each decoding method to generate 5,000 images using the captions from this dataset. We then calculate the FID scores between the images generated by each method and the ground-truth images by using the official API from 'pytorch_fid'. For the CLIP-Score evaluation, we utilize the CLIP-ViT-B pre-trained by OpenAI following [A]. We employ the official API from 'torchmetrics' to compute the CLIP-Score. Specifically, for each decoding method, we send each pair of generated image and its corresponding prompt to the API. After processing all pairs, the API can be used to compute the CLIP-Score. The evaluation follows the settings of the mainstream text-to-image generation works.
>
>     [A] Hessel J, Holtzman A, Forbes M, et al. Clipscore: A reference-free evaluation metric for image captioning[J]. arXiv preprint arXiv:2104.08718, 2021.
>
>
> 4. **The good FID scores may come from worse quality and better diversity.**
>
>     We would like to emphasize that the image quality of our method remains **comparable** to that of the original auto-regressive decoding. First, besides FID, we also report the CLIP-Score metric, which does not account for image diversity. As shown in Table 1, our CLIP-Scores are consistently around 31.3, indicating that our method does not pose much impact on image quality. Moreover, we employ the human preference score (HPSv2) to further compare the decoding methods. Please refer to the table below. The different decoding methods exhibit similar human preference scores, further demonstrating the preservation of image quality. Additionally, our SJD method preserves image diversity of standard auto-reregressive decoding, but not amplify the diversity. As shown in Figure 17, comparing SJD and standard AR, the images generated with the same prompt exhibit similar behaviors with SJD and standard AR. Given the maintenance of both image quality and diversity, the FID scores are similar for the original auto-regressive decoding and our SJD.
>
>     | Decoding Methods  |  HPSv2 |
>     |-------------|-------------|
>     | Auto-regression  | 0.2965 |
>     | Jacobi Decoding  | 0.3103 |
>     | Ours  | 0.3020 |
>
> 5. **About the number and size of tested models**
>
>     Actually, auto-regressive text-to-image generation with discrete image tokens is a relatively new research field, so there are few open-source models available. We have found *all the performant open-sourced models before the main submission deadline* (**Lumina-mGPT, Anole, LlamaGen-Stage1/Stage2, and Emu3**) to verify the effectiveness of our SJD.
>     Among these models, LlamaGen is the **smallest** with 775M parameters, while the others range from 7 to 8 billion parameters. The results for these models have been included in our paper:
>     - The results for Lumina-mGPT are presented in Table 1 and Table 2. The revision further adds Figure 5 and Figure 16.
>     - The results for Anole can be found in Table 1 and Table 2. The revision further adds Table 4, Table 5, and Figure 17.
>     - The results for LlamaGen-Stage1/Stage2 can be found in Table 3.
>     - The results for Emu3 can be found in Figure 11. Since Emu3 has the quantitative results on GenEval in its original paper, we perform the following experiment to demonstrate that SJD does not influence too much image quality. Please refer to the table below:
>
>     | Decoding Methods  |  GenEval |
>     |-------------|-------------|
>     | Auto-regression  | 0.64 |
>     | SJD  | 0.63 |
>
> 6. **About the perplexity**
>
>     We compare the perplexities between SJD and other decoding methods on Lumina-mGPT in Table 6 of the supplementary material of the revised paper. You can also refer to the table below. Since the perplexities are influenced by the sampling strategies, we report the perplexities under various top-$K$ sampling values. Given an identical $K$ value, the perplexities between our method and other decoding methods are close.
>
>     | Configuration | Top-10 Sampling | Top-100 Sampling | Top-2000 Sampling |
>     |-----------|:---------------:|:----------:|:-------------:|
>     | A. Lumina-mGPT | 7.31 | 43.37 | 204.06 |
>     | B. w. JD | 7.20 | 43.85 | 197.64 |
>     | C. w. SJD | 7.34 | 43.87 | 217.96 |
>     | D. w. SJD (ISP) | 7.26 | 44.03 | 199.70 |
>
> 7. **About the image quantity for FID on SJD and other compared methods**.
>
>     Our evaluation does **NOT** involve any tricks. For each decoding method, we have utilized the official API from 'pytorch_fid' to calculate the FID scores. Each method in these tables uses 5,000 image samples for calculating FID on the MSCOCO 2017 validation set. For further details regarding our evaluation, please kindly refer to our answer to question 3 above.

---

> > ### Author Response · Authors · 2024-11-20
> >
> > 8. **About the results on small auto-regressive text-to-image generation models**.
> >
> >     Please kindly refer to our answer to question 5 above. We tested on all available auto-regressive text-to-image generation models that are currently open-sourced.

---

> > ### Comment · Reviewer_EsPc · 2024-11-21
> > **Further doubts on perplexity and FID scores**
> >
> > 1) There seems to be little difference between different sampling results. There too little test images for FID which may be not suitable to evaluate the image quality.
> > 2) The FID score should be computed on COCO2014 test set with more than 30k images according to https://paperswithcode.com/sota/text-to-image-generation-on-coco.
> > 3) The authors's explanation of wierd JD performance is not persuasive because sampling accelaration is not relevant to dataset diversity and dataset size.
> > 4) Considering my original question3) and point 3) here, the authors should add experiments on small dataset and small models to demonstrate their generization ability.
> > #the authors claim that "The original Jacobi Decoding paper demonstrated its success on MNIST and CIFAR-10 class-conditioned image generation, but these datasets are much smaller, and the class-conditioned image generation on these datasets requires far less image diversity and details compared to current text-to-image generation. On these small datasets, models can use greedy decoding to generate images, so Jacobi Decoding works well. " #

---

> > > ### Author Response · Authors · 2024-11-22
> > >
> > > 1. **About little difference between different sampling results**
> > >
> > >     We acknowledge the observation that there seems to be little difference between different sampling results. However, the minimal difference between different sampling methods is due to the fact that ***our SJD incurs only minimal quality degradation***, which is the ***expected outcome***. Specifically, in Table 1 and Table 2, the minimal difference in CLIP-Scores demonstrates this fact. Moreover, the FID is not constant. For example, the FID scores for Lumina-mGPT and Anole are about 31 and 29, respectively, indicating that FID can indeed reflect the capabilities of different models.
> > >     We understand the concern that using 5,000 images for FID evaluation may not fully capture the nuances of image quality. ***We present the results on the COCO2014 validation set (30k images) in the next question.***
> > >
> > >
> > > 2. **About FIDs on COCO2014 validation set with 30k images**
> > >
> > >     To compute the FID score on the COCO2014 validation set with 30k images (a substantial quantity of testing images), we identified a ***small*** text-to-image generation model, ***LlamaGen-XL with only 775M parameters***. This model can generate images quickly and requires a maximum of only 5GB of GPU memory, making it even feasible for personal computers. To the best of our knowledge, it is currently the smallest open-sourced performant auto-regressive text-to-image generation model. The FID scores ***with classifier-free guidance as 3.0*** are in the table below. Moreover, reporting image quality on COCO2017 validation set (5k images) is also a common practice [a,b,c].
> > >
> > >
> > >     | Dataset | Configuration | Latency  $(\downarrow)$ | Acceleration Latency $(\uparrow)$ | Acceleration Step $(\uparrow)$ | FID $(\downarrow)$ |
> > >     |---------|---------------|------------------|------------------|---------|---------|
> > >     | COCOval2014 (30k images)  | LlamaGen-XL (775M) | 9.54s | $1.00 \times$          | $1.00 \times$         |  17.49  |
> > >     | COCOval2014 (30k images)   | LlamaGen-XL (775M) + **Ours** | 5.50s | **$\mathbf{1.73} \times$** | **$\mathbf{1.80} \times$**           |  17.78  |
> > >
> > >     [a] Kohler J, Pumarola A, Schönfeld E, et al. Imagine flash: Accelerating emu diffusion models with backward distillation[J]. arXiv preprint arXiv:2405.05224, 2024.
> > >
> > >     [b] Xu C, Song T, Feng W, et al. Accelerating Image Generation with Sub-Path Linear Approximation Model[C]. ECCV, 2024.
> > >
> > >     [c] Gu J, Zhai S, Zhang Y, et al. Boot: Data-free distillation of denoising diffusion models with bootstrapping[C]//ICML 2023 Workshop on Structured Probabilistic Inference {\&} Generative Modeling. 2023.

---

> ### Author Response · Authors · 2024-11-22
>
> 3. **Explanation on why Jacobi Decoding does not work for current text-to-image auto-regressive models:**
>
>     - **Difference between previous-generation auto-regressive models and the current-generation text-to-image auto-regressive models.** We would like to first clarify the significant differences between previous-generation auto-regressive image generation models such as PixelCNN (2016) and PixelCNN++ (2017) on MNIST and CIFAR-10, and current-generation auto-regressive text-to-image generation models such as LlamaGen (2024), Lumina-mGPT (2024), and Emu3 (2024). These differences can be summarized from two perspectives: ***(1) Data perspective:*** Current text-to-image generation tasks on ***diverse***, open-domain images are much ***more complex*** than previous image generation tasks on small-scale, low-diversity datasets like MNIST and CIFAR-10. Current text-to-image models require higher diversity and more detailed image generation. ***(2) Model perspective:*** Previous auto-regressive image generation models operate in pixel space and leverage ***small codebook sizes*** (e.g., PixelCNN has 256 codes), while current-generation autoregressive image generation models operate in latent space with ***large codebooks (8,000 to 16,000 codes)*** to sample from.
>
>     - **The current-generation text-to-image auto-regressive models require higher randomness in sampling, compared with previous-generation auto-regressive models.** Due to ***(1)*** and ***(2)***, previous-generation auto-regressive image generation models (e.g., PixelCNN, PixelCNN++) require ***limited randomness*** during inference, while current-generation auto-regressive text-to-image generation models (e.g., LlamaGen, Lumina-mGPT, Emu3) require ***higher randomness*** during inference. For example, LlamaGen leverages top-$1000$ sampling.
>
>     - **Jacobi Decoding, designed for the previous-generation auto-regressive models, is not compatible with high randomness in sampling.** At the time Jacobi Decoding was proposed, algorithms such as PixelCNN and PixelCNN++ required ***limited randomness*** for sampling. Designed for these algorithms, Jacobi Decoding effectively accelerates these models on datasets like MNIST and CIFAR. However, ***Jacobi Decoding*** is ***not*** compatible with ***highly random sampling*** such as top-$1000$ sampling in LlamaGen. This explains our motivation for why naive Jacobi Decoding does not work for auto-regressive text-to-image generation models and why it is necessary and non-trivial to design new algorithms to accelerate these models.
>
>     - **Summarization.** We summarize our above analysis as follows: Current auto-regressive text-to-image models are trained on ***large and diverse datasets*** with a ***large codebook size***. In these models, token sampling with high randomness (e.g., top-$2000$ sampling) is typically used to generate diverse images. With such highly random sampling, the original Jacobi Decoding is not effective because it is ***rare to encounter identical model outputs between two Jacobi iterations***. As evidenced by Figure 6, Jacobi Decoding can accelerate text-to-image generation with top-$1$ sampling (easy to get identical tokens between two iterations) but not with top-$2000$ sampling (hard to get identical tokens between two iterations).
>
>
>
> 4. **About testing small models for demonstrating the generization ability of SJD**
>
>     Please refer to question 2. We use a **small** text-to-image generation model, **LlamaGen-XL with only 775M parameters**. This model can generate images quickly and requires a maximum of only 5GB of GPU memory, making it even feasible for personal computers.

---

> ### Comment · Reviewer_EsPc · 2024-11-25
>
> Thank your for your replies!
>
> 1) I still doubt about the performance of JD. I think JD should at least accelerate 1.5 or faster. But the authors' result show that JD is invalid while their JD variant with randomness lead to 2.0 faster. (the authors argue that this phenomenon is caused by the size of codebook. If this is true, the variation should be contiguous rathor than no acceleration of original JD). If the authors could provide the acceleration rate variaiton according to codebook size, it will be persuasive.
> 2) Comparison to recent work Break the Sequential Dependency of LLM Inference Using
> LOOKAHEAD DECODING(icml 2024); This work also use SPECULATIVE sampling similar to this paper.
> 3) Comparisons to pixelCNN is strange in their reply. The authors should compare to Autoregesive models with small codebook size.
> In conclusion,  this works look not solid enough. It seems an application of jacobian decoding to text-to-image generation(using reject sampling to bring randomness into JD).
>
> I only have several papers about text-to-image with diffusion models and I hope AC to add a more prefessional reviewer. I doesn't prepare to give too much comments but the other two reviews also did not give prefesional comments on this paper(they only raise some corncerns on the experimental results). If I did not understand this paper well, forgive me (:

---

> > ### Author Response · Authors · 2024-11-30
> > **Response to Reviewer EsPc (1/2)**
> >
> > Thank you for your feedback. Below, we have carefully addressed each of your comments with detailed responses to ensure clarity in our work.
> >
> > 1. **Doubt on the acceleration rate of Jacobi Decoding**
> >
> >     - *Firstly*, we feel it necessary to clarify our complete logical flow for why Jacobi Decoding does not work well in auto-regressive text-to-image models. Considering only the diversity/size of the dataset (as mentioned in the reviewer's feedback on November 21) or only the size of the codebook (as mentioned in the reviewer's feedback on November 25) does not fully encompass our complete logical process. As we elaborated in our response dated November 22, the previous auto-regressive image generation models prior to 2017 and the contemporary auto-regressive text-to-image generation models exhibit significant differences in data diversity and model design (including codebook design). The utilization of more diverse data and more advanced model designs with **larger** codebook sizes both contribute to the increased need for **higher randomness in sampling** within current auto-regressive text-to-image models. However, Jacobi Decoding, which was designed based on the auto-regressive generation models proposed before 2017, is not compatible with **such high levels of randomness** in sampling because ***it is rare for identical model outputs to occur between two Jacobi iterations (we conduct an experiment in the following paragraph to demonstrate the existence of this phenomenon)***.
> >
> >     - *Secondly*, the reviewer suggested an experiment to show the continuous acceleration rate variation according to codebook size. Since all current auto-regressive text-to-image models have been *well-trained with a large codebook size (8k or 16k image tokens)*, directly altering the codebook could potentially cause the models to malfunction. Nevertheless, we design an experiment that bypasses direct codebook modification while demonstrating the **continuous acceleration rate variation according to the sampling strategy**. **We believe this experiment addresses the reviewer's concern**. Specifically, we select multiple $K$ values for top-$K$ sampling, ranging from $K = 1$ to $K = 10$, and test the acceleration rate of Jacobi Decoding for each $K$. As shown in the table below, we observe that ***the acceleration rate of Jacobi Decoding degrades gradually with the increase of the $K$ value, and Jacobi Decoding indeed achieves good acceleration results for small $K$ values***. This phenomenon demonstrates the **correctness** of our implementation of Jacobi Decoding. From the results, we also observe that auto-regressive text-to-image models struggle to generate images of good quality with *small* $K$ values, as indicated by the low HPSv2 scores. This is why Jacobi Decoding is not suitable for application on current auto-regressive text-to-image models.
> >
> >     | Token Sampling for Jacobi Decoding  | Average Step Acceleration | Average Steps |  Average Latency | HPSv2 |
> >     |---------|-----|-----|-----|-----|
> >     | top-1  | $2.46 \times$  | 958 | 40.21s | 0.1413 |
> >     | top-2  | $1.62 \times$  | 1456 | 61.63s | 0.2730 |
> >     | top-3  | $1.40 \times$  | 1688 | 71.47s | 0.2729 |
> >     | top-4  | $1.30 \times$  | 1805 | 76.19s | 0.2766  |
> >     | top-5  | $1.25 \times$  | 1890 | 79.59s | 0.2839  |
> >     | top-6  | $1.24 \times$  | 1909 | 80.59s | 0.2832  |
> >     | top-7  | $1.20 \times$  | 1963 | 82.92s |  0.2813 |
> >     | top-8  | $1.18 \times$  | 2001 | 84.46s | 0.2855 |
> >     | top-9  | $1.17 \times$  | 2022 | 85.33s | 0.2910 |
> >     | top-**10**  | $1.15 \times$  | 2041 | 85.99s |0.2960 |
> >     | top-**100**  | $1.06 \times$  | 2236 | 91.75s | 0.3089|
> >     | top-**2000**  | $1.04 \times$  | 2266 | 93.42s | **0.3103**|
> >
> > 2. **Comparison to Lookahead Decoding**
> >
> >
> >     We conducted a detailed comparison between Lookahead Decoding and our SJD using 30k images on Anole. The results are reported below:
> >
> >     | Decoding Methods  | Average Latency | Average Step Acceleration |  Average Steps |  Standard Deviation of Steps  | Minimal Steps | Maximal Steps |
> >     |---------|-----|-----|-----|-----|-----|-----|
> >     | Auto-regressive Decoding |  48.96s  |  1.00$\times$ | 1026  |  -  | 1026 | 1026 |
> >     | Lookahead Decoding |   45.97s |  1.12$\times$ | 942  |  104  | **214**   |1020  |
> >     | SJD |  **26.86s** | **1.97**$\times$  | **526**  |**50** |   **213**  |  **703**  |
> >
> >     According to the results, we find SJD is more stable than Lookahead Decoding. Specifically, although Lookahead Decoding and SJD can both achieve a maximal acceleration of 4.79$\times$ (with 214 and 213 steps respectively), Lookahead Decoding has a much larger standard deviation in the number of decoding steps (104 for Lookahead Decoding vs. 50 for SJD). Moreover, in the worst-case scenario, Lookahead Decoding has almost no acceleration, requiring 1020 decoding steps. The average acceleration of SJD is much better than Lookahead Decoding.

---

> > > ### Author Response · Authors · 2024-11-30
> > > **Response to Reviewer EsPc (2/2)**
> > >
> > > 3. **Correctness of the implementation of the original Jacobi Decoding and the solidness of this paper**
> > >
> > >     - The reviewer suggests comparing the acceleration rates of Jacobi Decoding under different codebook sizes, particularly with small codebook sizes. However, using the small codebook size (<256) of *current open-source auto-regressive text-to-image models* (which typically have thousands of image tokens) is impractical. Despite the difficulty in *directly* reducing the codebook size, we can *indirectly* reduce the number of available image tokens (the value of $K$ in top-$K$ sampling) during the random token sampling. The results show that **the acceleration rate of Jacobi Decoding decreases continuously with the increase of the top-$K$ value, demonstrating the correctness of our implementation**. **This experiment is detailed in our response to Question 1**.
> > >
> > >     - We would like to clarify that our work is not a simple application of Jacobi Decoding. Instead, we identify that Jacobi Decoding does not work well for current auto-regressive text-to-image models with high randomness in top-$K$ sampling, and propose the first approach that accelerates auto-regressive text-to-image generation without compromising the image quality of the generated images. This paper primarily emphasizes the significance of introducing a probabilistic criterion into the Jacobi iteration, and this forms our method, Speculative Jacobi Decoding. We have conducted extensive analysis and experiments (**at least 10 tables and 13 figures**) to validate the acceleration and visual preservation of our method. All the experiments requested by the reviewers to verify the solidness of the decoding methods used in this paper have been conducted and are presented in our responses. Additionally, our approach can be easily deployed on **four different open-source text-to-image auto-regressive models**, achieving acceleration across all. We promise to open-source the code.

---

### Official Review · Reviewer_3y1d · 2024-11-03

**Soundness:** 3
**Presentation:** 3
**Contribution:** 2
**Rating:** 6
**Confidence:** 3

**Summary:**

Recent auto-regressive models generate quality images but require thousands of next-token prediction steps during inference, which is significant in terms of inference speed. To address this, this paper proposes Speculative Jacobi Decoding (SJD), a training-free probabilistic parallel decoding algorithm. SJD introduces a probabilistic convergence criterion, allowing for sampling-based token decoding that maintains diversity and accelerates inference while overcoming the limitation of original Jacobi Decoding - deterministic, greedy decoding, which limits diversity to be incorporated into the image generation framework. This approach enables multiple token predictions per step in a probabilistic way (i.e., inserting randomness), reducing required steps while preserving image quality.

**Strengths:**

- Clear motivation about the limitation of existing performant auto-regressive T2I generation models and Jocobi decoding strategy.
- Achieving about 2 times faster inference latency with fewer iteration steps, with a small drop in image quality compared to original T2I backbones.
- This approach can be applied to multiple different auto-regressive T2I generation models.
- Provide several meaningful ablations to better understand the robustness of the proposed methods with varying scales of hyperparameters.

**Weaknesses:**

- The proposed approach requires over-computation in their Jacobi iteration. It computes conditional prediction scores for all tokens in the window but only accepts several earlier tokens by design - so resources used for computing later tokens are the first unaccepted tokens and later ones, even if some of them are technically accepted. This is inevitable since next-token prediction should be conditioned on all previous tokens that should be accepted (except for the beginning) and cannot allow some undesired holes of unaccepted tokens. While the approach proceeds these unaccepted tokens to new draft tokens for the next iterations, the effectiveness of this technique is not studied throughout the paper. At least, comparing the impact of this reuse of over-computed tokens with several token initializations with spatial priors (and random as well) would be meaningful.

- Initialization with spatial prior (ISP) doesn't show real acceleration in both terms of latency and steps compared to SJD w/o ISP. Figure 9 shows some different behavior, but the gap looks marginal in the main table (Table 1). I would say this is not appealing in terms of idea since this is from a random idea without strong grounding, less relevant to the overall story of this paper, and the performance is also less impactful.

- Analyses/ablations look incomplete. For Figures 6,7,8 and 9, I believe this should also be meaningful to provide quantitative performance of image quality rather than reporting only step compression rate. Reporting only the efficiency of the proposed approach does not give meaningful insight.

- The experimental results with Anole in Table 1 also look incomplete. I think more rows (i.e., B and C in Lumina-mGPT cases) are necessary for this table.

- (minor) typo: Each column presents --> Each row presents

**Questions:**

- The acceptance is determined if it improves prediction compared to the conditions of the previous Jacobi iteration. I wonder if the conditional probability given the draft tokens from the previous iteration is too low; it may have a chance to accept tokens even if it still has a very low conditional probability but a little bit higher than that, which may result in inaccurate token acceptance.

---

> ### Author Response · Authors · 2024-11-20
>
> We are grateful to the reviewer for their insightful review of our submission and for providing valuable suggestions. Below, we provide detailed responses to the questions and concerns raised.
>
> 1. **The effectiveness of reusing unaccepted tokens as new draft tokens.**
>
>     We conduct empirical analysis and find that reusing unaccepted tokens as draft tokens in the next iterations is critical to our method. To demonstrate this, we performed the following experiments, with quantitative and qualitative results presented in Figure 14 of the supplementary material of the revised paper:
>     When we deactivate the refinement (i.e., using newly initialized tokens to replace the unaccepted tokens as draft tokens in the next iteration), we observe that the model requires over two thousand forward passes to generate images, rather than one thousand. Although our token initializations with spatial prior (e.g., horizontal repeat) are slightly better than random token initialization, their acceleration is still significantly worse compared to directly refining the unaccepted tokens. You can also refer to the quantitative results below:
>     | Draft Token | Average Steps | Average Latency | Step Compression Ratio|
>     |-------|-------|-------|-------|
>     | Not Reusing Unaccepted Tokens + Random Token Initialization | 2356 | 107.65s | $1.00 \times$ |
>     | Not Reusing Unaccepted Tokens + Horizontal Repeat Token Initialization | 2281 | 95.85s | $1.03 \times$ |
>     | Reusing Unaccepted Tokens  | 1005 | 42.13s | $2.35 \times$ |
>
> 2. **About the initialization with spatial prior (ISP).**
>
>     We acknowledge that this design is not our major contribution. It is specifically designed to accelerate images in extreme cases, such as the images containing simple and repeating patterns. The acceleration ratios and generated images are showcased in our ablation studies (Figure 9, more than $3 \times$ step compression) and teaser (Figure 1, the ''Corgi dog'' case, $2.8 \times$ step compression). Although it has not achieved significant acceleration across all cases, this represents a preliminary exploration of using spatial priors for accelerating auto-regressive text-to-image generation, which would probably inspire future research to leverage spatial prior for the acceleration of auto-regressive text-to-image generation.
>
> 3. **Adding quantitative performance of image quality to Figures 6,7,8 and 9.**
>
>     Thank you for your feedback. We utilize the CLIP-Score and the human preference score (HPSv2) as the metrics for evaluating the visual quality of the images generated in our ablation studies. The results are presented in Table 7, Table 8, Table 9, and Table 10 of the supplementary material of the revised paper. From the results in these tables, we find that the change in visual quality is within a small range. Please refer to the tables below (We have bolded the adopted settings for qualitative results in Figure 15):
>
>     **Table 7: Visual quality for Figure 6:**
>
>     | Decoding Methods | Sampling | CLIP-Score |  HPSv2 |
>     |-------------|---------------|-------------|-------------|
>     | Auto-regression | Top-1 Sampling | 26.40 | 0.1976 |
>     | Auto-regression | Top-10 Sampling | 32.83 | 0.2950 |
>     | Auto-regression | Top-100 Sampling | 32.41 | 0.3020 |
>     | **Auto-regression** | **Top-2000 Sampling** | **32.00** | **0.2965** |
>     | Jacobi Decoding | Top-1 Sampling | 26.34 | 0.1413 |
>     | Jacobi Decoding | Top-10 Sampling | 32.75 | 0.2960 |
>     | Jacobi Decoding | Top-100 Sampling | 32.46 | 0.3089 |
>     | **Jacobi Decoding** | **Top-2000 Sampling** | **31.68** | **0.3103** |
>     | Ours | Top-1 Sampling | 26.16 | 0.1695 |
>     | Ours | Top-10 Sampling | 32.27 | 0.2942 |
>     | Ours | Top-100 Sampling | 32.65 | 0.2977 |
>     | **Ours** | **Top-2000 Sampling** | **31.83** | **0.3020** |
>
>     **Table 8: Visual quality for Figure 7:**
>
>     | Decoding Methods | Resolutions | CLIP-Score | HPSv2 |
>     |-------------|---------------|-------------|-------------|
>     | Auto-regression | 512 | 29.49 | 0.2503 |
>     | **Auto-regression** | **768** | **32.00** | **0.2965** |
>     | Auto-regression | 1024 | 31.41 | 0.2961 |
>     | Ours | 512 | 29.69 | 0.2558 |
>     | **Ours** | **768** | **31.83** | **0.3020** |
>     | Ours | 1024 | 31.11 | 0.2935 |
>
>     **Table 9: Visual quality for Figure 8:**
>
>     | Jacobi Window Size  | CLIP-Score | HPSv2 |
>     |-------------|-------------|-------------|
>     | 1 | 32.00 | 0.2965 |
>     | 4 | 31.91 | 0.3046 |
>     | **16** | **31.83** | **0.3020** |
>     | 32 | 31.55 | 0.3045 |
>
>     **Table 10: Visual quality for Figure 9 (evaluation on images with extremely simple patterns):**
>
>     | Token Initialization  | CLIP-Score | HPSv2 |
>     |-------------|-------------|-------------|
>     | Horizontal Sample | 31.52 | 0.2567 |
>     | Vertical Sample | 30.91 | 0.2622 |
>     | **Horizontal Repeat** | **31.17** | **0.2616** |
>     | Vertical Repeat | 31.15 | 0.2651 |
>     | Random | 31.37 | 0.2681 |

---

> ### Author Response · Authors · 2024-11-20
>
> 4. **Comparing SJD with ISP to other decoding methods on Anole.**
>
>     Thank you for your suggestion. We have conducted experiments comparing SJD with ISP to other decoding methods on Anole. Please refer to the tables below. These quantitative results are also presented as Table 4 and Table 5 in the supplementary material of the revised paper:
>
>     **COCO:**
>     | Configuration | Average Latency $(\downarrow)$ | Acceleration Latency $(\uparrow)$ | Acceleration Step $(\uparrow)$ | FID $(\downarrow)$ | CLIP-Score $(\uparrow)$ |
>     |---------|---------------|------------------|------------------|---------|----------------|
>     | A. Anole    | 48.96s | $1.00 \times$          | $1.00 \times$         | $28.87$   | $30.59$          |
>     | B. w. JD    | 47.60s | $1.03 \times$ | $1.06 \times$           | $29.34$   | $30.64$          |
>     | C. w. SJD    | 27.08s | $1.81 \times$           | $1.94 \times$           | $29.04$   | $30.54$          |
>     | D. SJD (ISP)    | 26.18s | **$\mathbf{1.87} \times$** | **$\mathbf{1.97} \times$**           | $29.14$   | $30.61$          |
>
>     **Parti:**
>     | Configuration | Average Latency $(\downarrow)$ | Acceleration Latency $(\uparrow)$ | Acceleration Step $(\uparrow)$ | CLIP-Score $(\uparrow)$ |
>     |---------|---------------|------------------|------------------|----------------|
>     | A. Anole    | 48.24s | $1.00 \times$          | $1.00 \times$          | $30.46$          |
>     | B. w. JD    | 44.65s | $1.08 \times$ | $1.14 \times$             | $30.57$          |
>     | C. w. SJD    | 26.77s | $1.80 \times$           | $2.00 \times$    | $30.55$          |
>     | D. SJD (ISP)    | 26.12s | **$\mathbf{1.92} \times$** | **$\mathbf{2.11} \times$**    | $30.48$          |
>
> 5. **Typo fix.**
>
>     Thank you for pointing out the typo, and we have corrected it.
>
> 6. **Discussion on inaccurate token acceptance when tokens have low conditional probability but slightly higher than previous iteration.**
>
>     We acknowledge that the mentioned corner case might happen in theory, where the draft token is sampled with very low conditional probability (confidence), yet the verification accepts it because the conditional probability (confidence) is higher than that of the previous iteration. However, we find that the tokens with low conditional probability (confidence) would be normally rejected by our method, meaning that **such corner case rarely happens**. We compute the statistics of the logarithm of the token probability for both auto-regressive decoding and our SJD. We find that the conditional probability (confidence) of image tokens accepted by our method exhibits similar statistics to those accepted by the original auto-regressive decoding. Consequently, our method generally does not mistakenly accept tokens with lower probabilities. The statistics are in the table below:
>
>     | Decoding Methods | Average of Logarithm of Token Probability | Standard Deviation of Logarithm of Token Probability |
>     |---------|---------------|------------------|
>     | Auto-regression    | -4.8950  |  2.3457  |
>     | Ours    |  -4.9007  |  2.3275     |
>
>     It is worth noting that even for standard auto-regressive decoding without SJD, the conditional probabilities of sampled tokens are not very high. This is because, in visual auto-regressive generation models, the conditional distribution for the next token is not an unimodal distribution (i.e., there could be more than one possibility for the next token given the previous ones). Moreover, visual generation requires diversity, so instead of sampling the next token with the highest probability, we add randomness by using top-$K$ sampling, further decreasing the conditional probability of the sampled tokens. Nonetheless, we did not observe a severe decrease in the conditional probability of the accepted tokens in SJD, compared with the sampled tokens in standard auto-regressive decoding.

---

### Official Review · Reviewer_8dQw · 2024-11-05

**Soundness:** 3
**Presentation:** 2
**Contribution:** 3
**Rating:** 6
**Confidence:** 4

**Summary:**

This paper proposes Speculative Jacobi Decoding (SJD) for accelerating the inference of auto-regressive (AR) image generation. SJD introduces a probabilistic acceptance criterion, using the ratio of the probability conditioned on the current draft tokens to that of the previous iteration for speculative verification, enabling faster convergence while preserving diversity in sampling-based decoding. This approach reduces the number of steps required, improving inference speed without compromising image quality.

**Strengths:**

1. This paper adapts speculative decoding to auto-regressive (AR) image generation, achieving substantial speed improvements with minimal impact on image quality, addressing a key efficiency challenge in this area.

2. The proposed method demonstrate performance gains on state-of-the-art models, including Lumina-mGPT and Anole, underscoring its applicability and effectiveness.

**Weaknesses:**

1. Need for analysis on image randomness in SJD. The randomness in SJD introduces variability in generated images, but there is limited analysis on the extent of this randomness. Unlike speculative decoding in the language domain, which offers some guarantees on reproducibility, SJD would benefit from theoretical bounds or deeper empirical analysis to clarify the extent of variation in its results.

2. Demonstrating similar speed-up and preservation of image quality on additional models, such as LlamaGen, would strengthen this paper.

3. Need for additional qualitative evaluation: Additional qualitative results, as in Fig. 5, comparing original images with those generated by the proposed method, along with steps and latency comparisons on Lumina-mGPT and Anole, would be beneficial. Particularly, a detailed discussion on representative failure cases and their causes is needed to clarify potential issues in image quality.

3. While this paper tends to emphasize step counts in terms of speed-up, latency is a more accurate metric for evaluating it. Therefore, using latency as the primary measure would enable a clearer comparision.

4. Improved clarification in Fig. 4 would enhance understanding of the proposed method.

**Questions:**

Please see the Weaknesses

---

> ### Author Response · Authors · 2024-11-20
>
> We thank the reviewer for taking the time to read our submission and provide constructive feedback. We will now respond to the highlighted questions and concerns.
>
> 1. **Analysis on the image randomness in SJD.**
>
>     Thank you for your insightful suggestion. Randomness is essitial for all visual generative models to achieve diversity in generated images. For auto-regressive text-to-image models without SJD, randomness is introduced from the sampling strategy (e.g., top-K sampling), resulting in diverse generated images. Our SJD is not designed to amplify such randomness, but to preserve this randomness. To compare the randomness of SJD and tranditional auto-regressive decoding, we have conducted a qualitative analysis to examine the image randomness, with the results presented in Figure 17 of the supplementary material of the revised paper.
>     *Firstly*, *SJD indeed introduce some randomness into image generation* (as represented by the random variable $r$ in our Equation 1). Consequently, the images generated using SJD cannot be exactly identical to those generated through auto-regression, even when the random seed is fixed. For example, in Figure 17, two images with the same $K$ (i.e., the value from top-$K$ sampling) and the same random seed (i.e., within the same column) cannot be exactly identical.
>     *Secondly*, comparing the images generated by SJD and standard auto-regressive decoding, *SJD has a negligible impact on the diversity of images*. For instance, when $K=2000$, comparing SJD and standard auto-regressive decoding, the images generated with the same prompt exhibit similar behaviors in the diversity of generated images.
>     *Thirdly*, *the diversity of generated images is mainly determined by the sampling strategies rather than the decoding methods*. In Figure 17, as the $K$ values in top-$K$ sampling increase, the details related to textures, colors, and local structures consistently become more diverse in SJD and standard auto-regressive decoding.
>
> 2. **The quantitative results of LlamaGen**.
>
>     The results for LlamaGen are presented in the table below. These results have been included in Table 3 of our supplementary material.
>
>     | Dataset | Configuration | Acceleration Latency $(\uparrow)$ | Acceleration Step $(\uparrow)$ | FID $(\downarrow)$ | CLIP-Score $(\uparrow)$ |
>     |---------|---------------|------------------|------------------|---------|----------------|
>     | COCOval2017    | LlamaGen-stage1 | $1.00 \times$          | $1.00 \times$         | $28.54$   | $30.87$          |
>     | COCOval2017    | LlamaGen-stage1 + **Ours** | **$\mathbf{1.56} \times$** | **$\mathbf{1.63} \times$**           | $29.00$   | $30.82$          |
>     | COCOval2017    | LlamaGen-stage2 | $1.00 \times$           | $1.00 \times$           | $56.21$   | $28.26$          |
>     | COCOval2017    | LlamaGen-stage2 + **Ours** | **$\mathbf{1.54} \times$** | **$\mathbf{1.63} \times$**           | $57.02$   | $28.33$          |
>     | Parti-Prompt   | LlamaGen-stage1 | $1.00 \times$           | $1.00\times$           | -       | $30.22$          |
>     | Parti-Prompt   | LlamaGen-stage1 + **Ours** | **$\mathbf{1.57} \times$** | **$\mathbf{1.73} \times$**           | -       | $30.29$          |
>     | Parti-Prompt   | LlamaGen-stage2 | $1.00 \times$           | $1.00 \times$           | -       | $28.14$          |
>     | Parti-Prompt   | LlamaGen-stage2 + **Ours** | **$\mathbf{1.62} \times$** | **$\mathbf{1.69} \times$**           | -       | $28.16$          |
>
>     Note that the FID score for LlamaGen-stage2 is high because this model typically generates images with various styles that are distant from the distribution of COCO images. This is a property of the original LlamaGen-stage2 model [A] which is not related to our SJD algorithm.
>
>     [A] Sun P, Jiang Y, Chen S, et al. Autoregressive Model Beats Diffusion: Llama for Scalable Image Generation[J]. arXiv preprint arXiv:2406.06525, 2024.
>
>
> 3. **Additional qualitative evaluation**
>
>     We have included additional qualitative results for Lumina-mGPT and Anole in the supplementary material of the revised paper, specifically in Figure 15 and Figure 16, and we report both the steps and latency. According to the reported latency and step compression in these figures, our SJD outperforms other decoding methods while maintaining visual quality.
>
> 4. **Discussion on failure cases and their causes.**
>
>     Thank you for your valuable suggestion. We observe that the SJD method can introduce artifacts when generating images with intricate details, as shown in Figure 18 of the supplementary material of the revised paper. The pre-trained auto-regressive model is not sufficiently robust to handle such complex images. Consequently, it may mistakenly accept a sequence of draft tokens that contain artifacts.

---

> > ### Author Response · Authors · 2024-11-20
> >
> > 5. **About the latency evaluation in method comparison.**
> >
> >     Thank you for your feedback. We have presented the latency in Table 1, Table 2, Table 4 and Table 5 of the revised manuscript. You may also refer to the tables below. Specifically, we have included the actual latency and the latency acceleration in these tables. Additionally, Figure 15 in the supplementary material of the revised paper now provides a detailed comparison of the actual latency when SJD is compared with other decoding methods across various prompt cases.
> >     **COCO:**
> >
> >     | Configuration | Average Latency $(\downarrow)$ | Acceleration Latency $(\uparrow)$ | Acceleration Step $(\uparrow)$ | FID $(\downarrow)$ | CLIP-Score $(\uparrow)$ |
> >     |---------|---------------|------------------|------------------|---------|----------------|
> >     | A. Lumina-mGPT    | 87.23s | $1.00 \times$          | $1.00 \times$         | $30.76$   | $31.29$          |
> >     | B. w. JD    | 85.20s | $1.02 \times$ | $1.04 \times$           | $30.66$   | $31.38$          |
> >     | C. w. SJD    | 42.73s | $2.04 \times$           | $2.22 \times$           | $30.85$   | $31.35$          |
> >     | D. SJD (ISP)    | 42.49s | **$\mathbf{2.05} \times$** | **$\mathbf{2.23} \times$**           | $31.13$   | $31.33$          |
> >
> >     | Configuration | Average Latency $(\downarrow)$ | Acceleration Latency $(\uparrow)$ | Acceleration Step $(\uparrow)$ | FID $(\downarrow)$ | CLIP-Score $(\uparrow)$ |
> >     |---------|---------------|------------------|------------------|---------|----------------|
> >     | A. Anole    | 48.96s | $1.00 \times$          | $1.00 \times$         | $28.87$   | $30.59$          |
> >     | B. w. JD    | 47.60s | $1.03 \times$ | $1.06 \times$           | $29.34$   | $30.64$          |
> >     | C. w. SJD    | 27.08s | $1.81 \times$           | $1.94 \times$           | $29.04$   | $30.54$          |
> >     | D. SJD (ISP)    | 26.18s | **$\mathbf{1.87} \times$** | **$\mathbf{1.97} \times$**           | $29.14$   | $30.61$          |
> >
> >     **Parti:**
> >
> >     | Configuration | Average Latency $(\downarrow)$ | Acceleration Latency $(\uparrow)$ | Acceleration Step $(\uparrow)$ | CLIP-Score $(\uparrow)$ |
> >     |---------|---------------|------------------|------------------|----------------|
> >     | A. Lumina-mGPT    | 100.69s | $1.00 \times$          | $1.00 \times$          | $32.13$          |
> >     | B. w. JD    | 100.00s | $1.01 \times$ | $1.04 \times$             | $32.17$          |
> >     | C. w. SJD    | 47.52s | $2.12 \times$           | $2.26 \times$    | $32.13$          |
> >     | D. SJD (ISP)    | 47.35s | **$\mathbf{2.13} \times$** | **$\mathbf{2.28} \times$**    | $32.06$          |
> >
> >     | Configuration | Average Latency $(\downarrow)$ | Acceleration Latency $(\uparrow)$ | Acceleration Step $(\uparrow)$ | CLIP-Score $(\uparrow)$ |
> >     |---------|---------------|------------------|------------------|----------------|
> >     | A. Anole    | 48.24s | $1.00 \times$          | $1.00 \times$          | $30.46$          |
> >     | B. w. JD    | 44.65s | $1.08 \times$ | $1.14 \times$             | $30.57$          |
> >     | C. w. SJD    | 26.77s | $1.80 \times$           | $2.00 \times$    | $30.55$          |
> >     | D. SJD (ISP)    | 26.12s | **$\mathbf{1.92} \times$** | **$\mathbf{2.11} \times$**    | $30.48$          |
> >
> > 6. **Improved clarification in Figure 4**
> >
> >     Thank you for your suggestion. We have modified Figure 4 to improve its readability and clarity.

---

> ### Comment · Reviewer_8dQw · 2024-12-03
>
> Thank you for conducting additional experiments and providing a detailed response.
>
> I still believe that this paper requires further analysis or discussion regarding the randomness in speculative jacobi decoding for visual AR models. However, as this work represents an early attempt to apply speculative decoding to visual AR models, including the current version of the discussion on the randomness in the main paper could be appropriate and provide valuable insights for future researchers.
> Considering the strengths of this work and its potential impact, I will maintain my score as borderline accept.

---

> > ### Author Response · Authors · 2024-12-03
> >
> > Dear Reviewer 8dQw,
> >
> > We sincerely appreciate your thorough review, thoughtful suggestions, and kind acknowledgment of our work. Your feedback was very useful and helped us improve the quality of our submission. Your suggested discussions have been incorporated into our revised paper: (1) Analysis on the image randomness in SJD (Figure 17, Section I); (2) The quantitative results of LlamaGen (Table 3, Section D); (3) Additional qualitative evaluation (Figure 15 and Figure 16, Section B); (4) Discussion on failure cases and their causes (Figure 18, Section J). We hope this work will inspire future research in accelerating auto-regressive visual content generation.
> >
> > Best regards,
> >
> > The Authors of Submission 1119

---

### Meta-Review · Area_Chair_K6i2 · 2024-12-25

**Metareview:**

This paper proposes Speculative Jacobi Decoding, a training-free probabilistic parallel decoding algorithm for accelerating auto-regressive text-to-image generation. The core novelty lies in introducing a probabilistic convergence criterion, which allows for faster decoding without sacrificing visual quality or diversity. The authors conduct multiple experiments demonstrating speed-ups across multiple state-of-the-art models while maintaining image quality. Overall, this work makes a meaningful contribution to auto-regressive visual generative models, and thus I recommend acceptance.

**Additional Comments On Reviewer Discussion:**

The reviewers raised several questions, primarily regarding additional experimental results. In response, the authors conducted comprehensive experiments that effectively addressed most of these concerns. As one of the first attempts to explore speculative decoding in auto-regressive visual generative models, the contributions of this paper outweigh its limitations.

---

### Decision · Program_Chairs · 2025-01-22

Accept (Poster)